# Assessing population structure and body condition to inform conservation strategies for a small isolated Asian elephant (*Elephas maximus*) population in southwest China

**Yakuan Sun**[1©], **Ying Chen**[2©], **Juan José Díaz-Sacco**[1©], **Kun Shi**[1,3©]*

**1** School of Ecology and Nature Conservation, Beijing Forestry University, Beijing, China, **2** School of Biological Science, The University of Hong Kong, Hong Kong, China, **3** Eco-Bridge Continental, Beijing, China

© These authors contributed equally to this work.

* kunshi@bjfu.edu.cn

**Data Availability Statement:** All relevant data are within the manuscript and its Supporting Information files.

## Abstract

The Asian elephant (*Elephas maximus*) population in Nangunhe National Nature Reserve in China represents a unique evolutionary branch that has been isolated for more than twenty years from neighboring populations in Myanmar. The scarcity of information on population structure, sex ratio, and body condition makes it difficult to develop effective conservation measures for this elephant population. Twelve individuals were identified from 3,860 valid elephant images obtained from February to June 2018 (5,942 sampling effort nights) at 52 camera sites. Three adult females, three adult males, one subadult male, two juvenile females, two juvenile males and one male calf were identified. The ratio of adult females to adult males was 1:1, and the ratio of reproductive ability was 1:0.67, indicating the scarcity of reproductive females as an important limiting factor to population growth. A population density of 5.32 ± 1.56 elephants/100 km$^2$ was estimated using Spatially Explicit Capture Recapture (SECR) models. The health condition of this elephant population was assessed using an 11-point scale of Body Condition Scoring (BCS). The average BCS was 5.75 (n = 12, range 2–9), with adult females scoring lower than adult males. This isolated population is extremely small and has an inverted pyramid age structure and therefore is at a high risk of extinction. We propose three plans to improve the survival of this population: improving the quality and quantity of food resources, removing fencing and establishing corridors between the east and wet parts of Nangunhe reserve.

## Introduction

The Asian elephant (*Elephas maximus*), as the largest terrestrial mammal in Asia, is considered an ecosystem engineer because it can modify its habitat and aid in seed dispersal [1, 2]. As an umbrella species, its conservation contributes to the preservation of biodiversity in the tropical moist and tropical dry broadleaf forests of Southeast and South Asia [3–5]. It is also considered

**Funding:** Funding for this work was provided by National Forest and Grassland Administration, http://www.forestry.gov.cn, China under the Project of Asian Elephant Habitat Assessment and Habitat Maintenance Pilot, with effective support by the Yunnan Forest and Grassland Administration. The funders had no role in study design, data collection and analysis, decision to publish, or preparation of the manuscript.

**Competing interests:** The authors have declared that no competing interests exist.

a flagship species that can attract interest in the protection of biodiversity [6]. It is a Class I Key Protected Species in China and listed as Endangered under the IUCN Red List [7, 8]. China sustains a population of around 300 wild Asian elephants, divided into seven geographic populations [9, 10]. Most of these populations are located in Mengyang, Mengla and Shangyong areas of Xishuangbanna National Nature Reserve, while there are also a few populations in Simao, Jiangcheng and Lancang counties of Pu'er city and one small isolated population in the Nangunhe National Nature Reserve (NNNR) of Lincang city [9, 11].

The conventional methods used for tracking Asian elephants in dense forest are difficult to accomplish and usually fail to provide accurate estimation of population size and age structure [12, 13]. Although it is relatively easy to detect signs of elephant presence, such as dung or footprints, during sign surveys [14], estimates of the number of individual Asian elephants derived from signs are highly inaccurate and unreliable [15]. Individuals can be identified using non-invasive DNA methods, but this requires relatively fresh fecal samples and the lab costs can be quite high. Population size can only be estimated from DNA using genetic capture-mark-recapture methods, which can be time and labor intensive [16, 17]. Furthermore, since elephant calves have lower defecation rates than adults, this method can underestimate the number of young individuals in a population [18].

Infrared-triggered camera technology can make up for the above-mentioned disadvantages [19, 20]. Cameras cause minimal disturbance to animals and their habitat, can be used continuously for long periods of time, can withstand harsh weather conditions, and are relatively cheap [19, 21, 22]. They are widely used to conduct baseline surveys of protected species [23, 24], to monitor endangered wildlife [25], to study habitat use [26], to observe animal behavior [27], and to study the mechanisms of coexistence in sympatric species [28, 29]. Moreover, infrared cameras can also be used to estimate population size and density in conjunction with Spatially Explicit Capture-Recapture models (SECR) [30–33]. This technology has been used extensively to estimate the population density of multiple species, especially those possessing easily observed markings, such as the snow leopard (*Panthera uncia*), the leopard cat (*Prionailurus bengalensis*), the Sumatran tiger (*Panthera tigris* ssp. Sumatrae), and the serval (*Leptailurus serval*) [33–36]. For species without natural markings, individual identification can still be conducted using morphological features. A number of studies have identified individual elephants in this way by directly photographing them or by obtaining images from infrared cameras [17, 37–39].

Effective conservation strategies for small, threatened populations need detailed baseline information on both the population size and its habitat. Population structure and density estimation are other essential features that can be used to study the ecology of wildlife populations [40, 41] while, the Body Condition Scoring (BCS) is an index of an animal's health that reflects habitat quality [42]. It can be used in conjunction with population structure and density estimation to obtain a more accurate picture of the health of a population. Furthermore, it can be used for both longitudinal and cross-sectional studies, is comparatively inexpensive, and is easy to use [43]. While BCS is individual-based, it is most meaningful when applied to a population as an early indicator of the impact of management actions on the average health of the population [43, 44]. For Asian elephants, the BCS obtained through photographs can be as accurate as an ultrasonic subcutaneous fat measurement, thus providing a safer and non-invasive method to evaluate the health condition of wild populations [43, 45].

In China, population studies on Asian elephants have mainly been concentrated in Xishuangbanna National Nature Reserve [17, 37]. These studies were limited to population size and none have assessed the health condition of the population [13, 46]. Although it is a unique evolutionary unit in China, the isolated Nangunhe National Nature Reserve population has been poorly studied. The lack of scientific data and information on its population

structure, genetic relatedness, and health status prevent the development of efficient conservation strategies. Additionally, this population is facing threats from habitat loss, fragmentation and illegal hunting [47–49]. An accurate understanding of the current situation of the elephant population in NNNR, especially its structure and health condition, is key to establishing a baseline for the development of more effective protection and management schemes.

In this study, our aim was to determine the size, density, demographic structure and health condition of the NNNR Asian elephant population using SECR and BCS with data from infrared camera trapping. We sought to understand which factors might have affected the population size in recent years in order to develop targeted and effective protection strategies.

## Material and methods

### Ethics statement

Both National Forestry and Grassland Administration of China and Yunnan Forestry and Grassland Administration reviewed all sampling procedures and approved permits for the work conducted in NNNR. Non-invasive methods were applied and approval from an Institutional Animal Care and Use Committee or equivalent animal ethics committee was therefore not required.

### Study area

The study area is located in the western part of the Nangunhe National Nature Reserve, Yunnan, where elephants are found [47, 48] (Fig 1). It occupies 70.82 km$^2$ at latitude 23˚13'- 23˚19'N, longitude 98˚54'- 99˚05'E, and the altitude ranges from 510 m to 1747 m. Rainfall is highly seasonal, with a dry season between November and April, and a rainy season between May and October. Mean precipitation is 351.3 mm in the dry season and 1983.3 mm in the rainy season, while the temperature varies at different altitudes. The average monthly temperature is hottest in July, varying between 12.3–25.5˚C and it is coldest in January, with values between 2.8–14.3˚C [50]. Vegetation and climate also vary across elevation zones. Tropical forest and tropical monsoon forest are located at altitudes below 600 m, middle elevation ranges are dominated by subtropical monsoon evergreen broad-leaf forests, and evergreen broadleaf forests are found above 1700 m [48]. Bamboo forest, shrub, and grasslands are scattered throughout the area. Deforestation along the China and Myanmar border has disrupted the connectivity between elephant populations that are found in both countries [47]. An iron fence in the southern part of the reserve is used to protect the surrounding villages from wildlife (Fig 1).

### Camera trap set-up

We carried out the survey for four months between February and May 2018 using automatically triggered camera traps (Ltl Acorn 6210). 52 cameras stations recorded for 43–129 camera-trap nights totaling 5,942 trap nights (114.27±19.83) (Fig 1). All camera stations were located in either evergreen broadleaf forest, broadleaf and bamboo forest, bamboo forest, pine forest or nearby brush forest, which represents the preferred habitat of elephants in NNNR. We systematically deployed camera traps along trails and spaces where there were signs of elephants, including lying sites, foraging sites, footprints, dung, and urine marks. We attempted to deploy cameras throughout the reserve according to a 1 km$^2$ grid. We increased the number of cameras in the grid cells where many elephant signs were found, but always with a minimum of 200 m between any two cameras. We avoided the northwestern part of the reserve as the topography is rough, with steep cliffs of up to 1747 meters high, and has no records of elephants [51, 52].

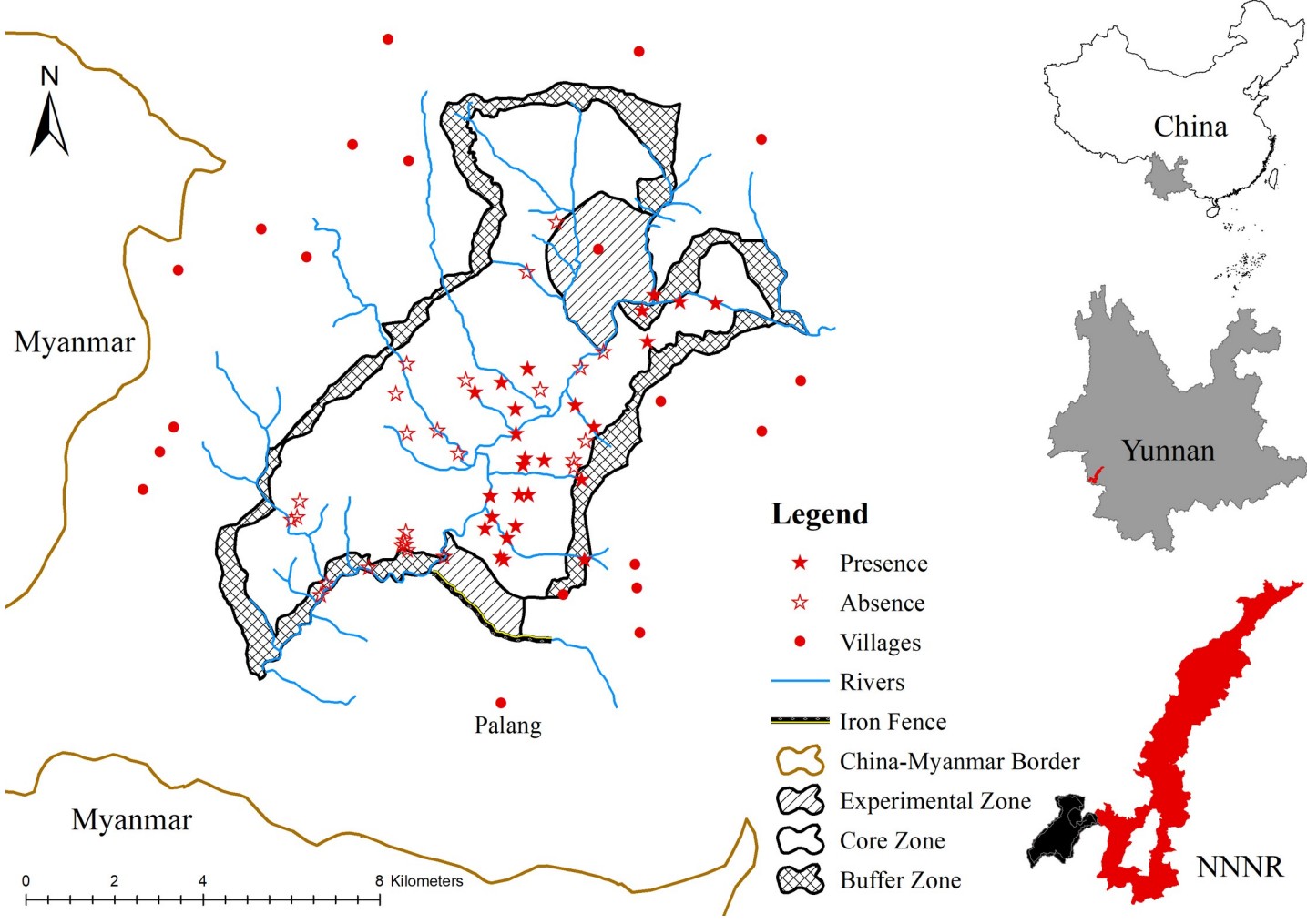

**Fig 1. Map of study area in NNNR showing the camera sites with or without elephant detections.**

Cameras were strapped onto trees at approximately 1.5 m above the ground with the sensor parallel to the ground to maximize the extension of the detection zone. The positions of all the camera sites were recorded by GPS. To avoid blank images caused by sunlight, the cameras were never placed facing east. Camera traps were hidden as much as possible to reduce the disturbance to wildlife and the probability of being stolen. Camera traps were set to operate 24 hours per day and programmed to take three photographs and one 12-second video in a sequence without delay. Cameras registered the date and time for each exposure. All camera traps were left operating continuously for a maximum of four months. SD cards and batteries were replaced every month. We recorded the time of installation and retrieval of each camera and calculated the total duration of sampling.

## Individual identification

The images obtained from the camera traps were categorized into independent events, which were defined as "sequences" of adjacent images. For one elephant unit (defined as all or the maximum number of individuals in a group), the sequence included all the images from the first individual entering the field to the last one exiting it, making it possible to identify

**Table 1. Morphological characteristics used for individual identification of elephants photographed in NNNR, between February and May 2018.**

| Characteristics | Categories | Description |
| --- | --- | --- |
| Age | Adult (>15years) | With the highest shoulder height |
| | Subadult (5-15years) | Shoulder height reaches an adult's eyes |
| | Juvenile (1-5years) | Shoulder height reaches half the height of an adult |
| | Calf (0-1year) | Below an adult's knee |
| Sex | Female | Without obvious tusks but with prominent breasts, belly swelling or the presence of nursing offspring |
| | Male | With visible testicles or penis, and/or obvious tusks |
| Facial bones | Wisdom tumor | Size |
| | Bones between the eyes | Extent of prominence |
| | Upper margin of eyes | Extent of depression |
| Tusks | Length | Very short/short/long compared with trunk |
| | Thickness | Thick/normal/slender |
| | Shape | Straight/bent |
| | Crack | Yes/no |
| Ear | Upper edge | Fold/unfold |
| | Lobe | V-acute/U-rounded |
| | Lobe tear | Yes/no |
| | Ear hole | Yes/no |
| Tail brush | Standard | Present/both sides/both-continuous |
| | Nonstandard | Absent/single side/both-discontinuous |
| Others | Old scars and injuries | - |

different individuals based on distinct features. For solitary individuals, a sequence was a series of captured images of the same individual within a continuous time period [53]. Only images from which age and sex classification was possible were used for individual identification [39, 53]. Animals were classified as either adult, subadult, juvenile or calf using approximate shoulder height as recommended by previous studies, and sex was mainly determined by secondary sexual characteristics [54–56] (Table 1). We defined the age structure as the ratio of adults, subadults, juveniles and calves. Reproductive ability was defined as the ratio of age structure between adult and juvenile females [39], due to the absence of subadult and calf females in this reserve. Male elephants living completely or mostly outside the herd were classified as solitary males [12]. Based on individual events, a capture-history event database was established to record the individuals and their characteristics. Individuals were identified based on morphological features including facial bone structures, ear shape and size, tusks, presence of back and tail hair, old scars and injuries, as well as physical characteristics related to sex and age classes [17, 37, 38, 57] (Table 1).

## Density estimation

We used a spatially explicit capture-recapture models (SECR) to estimate density of elephants in NNNR. The population is small and isolated and without any migrants, thus conforming to the "closed population" assumption of SECR [58]. The model accounts for movement and detect ability when estimating density by combining a state model and an observation model, which together describe how the observed data were generated [58]. The state model describes statistically the mechanisms determining the distribution of the animals in the study area, while the observation model quantifies the probabilities of detection or capture, given the surveyed region, the animals' locations and characteristics during different sessions [59]. SECR models can be described as generalized linear mixed models that assume that each individual $i$

in the population has its own unobserved activity center $S_i$, and that all activity centers $S_i \cdots S_N$ are distributed across the study area [60, 61].

The encounter rate of an individual $i$ with a given trap $j$ located at $X_j$ is a monotonically decreasing function of the distance from $S_i$ to $j$. $Y_{ij}$, the number of times individual $i$ is detected by trap $j$ during a sampling occasion, is a random variable following a Poisson distribution and with mean $\lambda_{ij}$ [62]:

$$Y_{ij} \sim \text{Poisson}(\lambda_{ij})$$

For an elephant $i$ captured by a trap $j$, the model assumes a log-linear form [63]:

$$\text{Log}(\lambda_{ij}) = \text{Log}(\lambda_0) - \left(\frac{1}{2\sigma^2}\right)\|S_i - X_j\|$$

Here, $\lambda_0$ is the baseline encounter probability, the function $\|S_i - X_j\|$ is the distance between the activity center and the trapping station, and $\sigma$ is the Gaussian scale parameter for the distance function between activity centers and trap locations; assuming a bivariate normal model of space use, sigma can be used to calculate a 95% activity area radius.

SECR models estimate the density of animal activity centers in an area large enough that animals residing beyond it have a negligible chance of being detected [58]. The maximum distance between the east and west boundaries of elephant distribution was 10 km, and so we defined our study area as extending 5 km beyond all camera stations, corresponding to an area of 255.8 km$^2$ [64].

SECR 3.1.5 package in R 3.6.1 was used for density estimation. Data were prepared as an R object of class "capthist". This object included both the capture data and the detector layout. The file named "capture histories" contained data on individual identified at each sampling occasion from the specific detector (camera). The detector file contained data on cameras' ID, and x-y coordinates. Five different sessions, corresponding to each of the months and one corresponding to the whole survey period, were used to run the model to see the estimation accuracy under different detection efforts.

## Body condition scoring

We established a database of each identified individual and the images with the best angle of each individual elephant were used to score the body condition. Compared to previous studies, our target population size was small, and the sample size was limited, therefore we used a simple method which can rapidly and more easily be applied to wild elephant populations. In this study, we applied a BCS method that focuses on six physical characteristics and ranks the elephant´s body condition according to an 11-point scale [42, 43]. Five of the clearest images for each physical characteristic were selected to assess the body condition. Thirty images were used to calculate the comprehensive scores for each individual. This scoring method is based on the extent of visibility of depressions around bone structures. Depressions around bones become visible as an animal loses its subcutaneous fat deposits and muscles in the region concerned, thereby making bones appear more prominent. Skull, pectoral girdles in shoulders, vertebral column, ribs and pelvic girdles are prominent bone structures. Thick rolls of skin that fold below the neck can accumulate fat, and thus, also represent a criterion of good body condition. We created a reference scale of 5 scored elephants with rankings of 1, 3, 5, 7, 9, with numerically higher values corresponding to better body condition (Table 2). However, the range of scores extended from 0 to 10 due to the possibility of greater variation. For example, if all the ribs of an elephant were visible, it meant that this individual presented a worse body condition than 1 and thus, it would be assigned 0. If an individual presented a fatter condition than 9 and had very thick rolls of skin fold below the neck, it would be assigned a score of 10.

**Table 2. Diagnostic characteristics pertaining to scores in photographic scale.**

| Score | Characteristics |
|---|---|
| 1 | Ribs from shoulder to pelvis are visible, some ribs prominent (spaces in between sunken in) |
| 3 | Some ribs visible (spaces in between not sunken in), shoulder and pelvic girdles prominent |
| 5 | Ribs not visible, shoulder and pelvic girdles visible |
| 7 | Backbone visible as a ridge, shoulder and pelvic girdles not visible |
| 9 | Backbone not visible or difficult to differentiate and pelvic bones not visible |

Scores ranging from 0–3 were considered poor, 4–7 were considered medium and 8–10 were considered good [44]. Pearson's correlation coefficient ($r$) was used to study the relationship between BCS and three covariates: sex, age, and whether an individual was solitary or in a herd.

## Results

### Population size and density

70,902 images were obtained, of which 7.34% (5,207) contained Asian elephants. There was a total of 134 independent events, including 40 events of elephants in a herd and 94 events of solitary elephants. Excluding the image data that could not be used for individual identification, there were 3,860 valid Asian elephant images belonging to 85 independent events (26 events of individuals in a herd and 59 events of individuals). Generally, all herds and individuals had similar frequencies of recapture histories during the study period, particularly herds. However, solitary elephants were image-captured with a minor difference: a minimum of 16 capture times for AE12 compared to a maximum of 25 capture times for AE10 (Table 3).

A total of 12 individuals were identified among these valid independent events (Fig 2; S1 Dataset). The elephant population size in each session was consistent, except in February. Moreover, the population size estimated by SECR was 12.00 ± 0.71 (95%CI 12.00,12.11; Table 4). The estimated density by SECR was 5.32 ± 1.56 elephants/100km$^2$ (95%CI 3.02,9.36). April had the lowest monthly density estimation at 6.07 ± 1.79 elephants/100km$^2$, while May had the highest density estimation at 24.43 ± 7.20 elephants/100km$^2$. The movement parameter σ was twice as large in April (6643 m ± 2356 m) than in May (3771 m ± 1471 m).

**Table 3. Number of capture events per individual per month.**

| Individual No. | Age | Gender | Tusk | Type | Number of Capture events | | | | |
|---|---|---|---|---|---|---|---|---|---|
| | | | | | Feb. | Mar. | Apr. | May | Total |
| AE01 | Juvenile | Male | Yes | Herd | 2 | 6 | 5 | 8 | 21 |
| AE02 | Adult | Female | No | Herd | 3 | 5 | 6 | 6 | 20 |
| AE03 | Adult | Female | No | Herd | 3 | 5 | 6 | 7 | 21 |
| AE04 | Juvenile | Female | No | Herd | 2 | 6 | 5 | 7 | 20 |
| AE05 | Calf | Male | No | Herd | 3 | 6 | 8 | 5 | 22 |
| AE06 | Adult | Female | No | Herd | 4 | 6 | 8 | 5 | 23 |
| AE07 | Juvenile | Female | No | Herd | 4 | 6 | 9 | 6 | 25 |
| AE08 | Juvenile | Male | Yes | Herd | 4 | 7 | 6 | 7 | 24 |
| AE09 | Sub-adult | Male | No | Solitary | 3 | 9 | 2 | 9 | 23 |
| AE10 | Adult | Male | Yes | Solitary | 2 | 3 | 13 | 7 | 25 |
| AE11 | Adult | Male | No | Solitary | 3 | 5 | 8 | 5 | 21 |
| AE12 | Adult | Male | No | Solitary | 2 | 6 | 1 | 7 | 16 |
| Total | - | - | - | - | 35 | 70 | 77 | 79 | 261 |

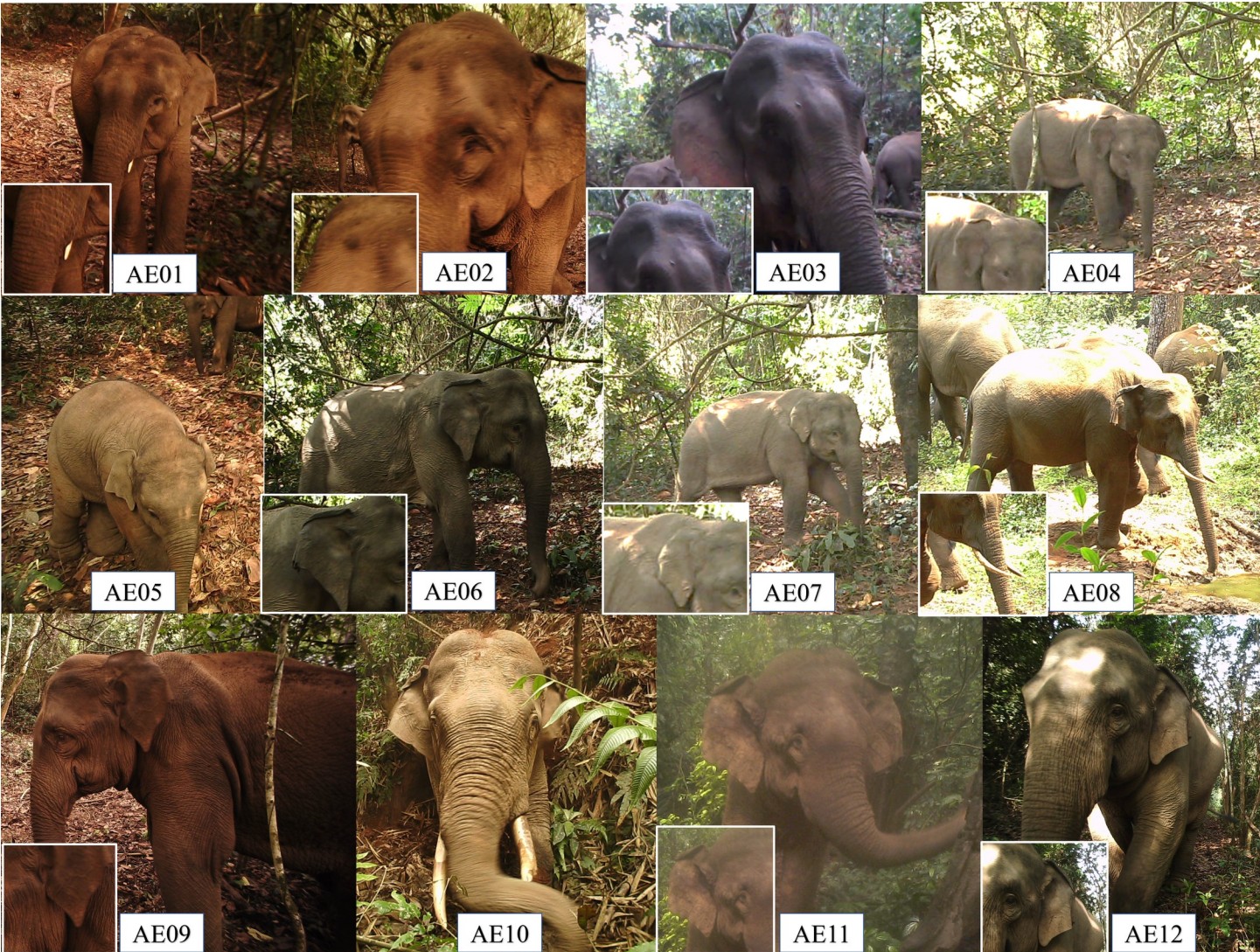

**Fig 2. Asian elephants individually identified in NNNR, China.** Each individual's most distinguished characteristic was emphasized, except for the easily identified AE05 and AE10, the only calf and adult male with tusks respectively.

## Demographic structure

We identified an inverted pyramid age structure for this population: three adult females, three adult males, one subadult male, two juvenile females, two juvenile males and one male calf, with a sex ratio of 1:1.4 using females as a reference (Table 4). The three adult males comprised

**Table 4. Population density estimation using SECR models for five sessions.**

| Sessions | Occasions | Density | | | $\lambda_0$ | | sigma | | Population size | | |
|---|---|---|---|---|---|---|---|---|---|---|---|
| | | estimate | SE.estimate | 95% CI | estimate | SE.estimate | estimate | SE.estimate | estimate | SE.estimate | 95% CI |
| February | 11 | 10.71 | 3.31 | 5.91–19.38 | 0.15 | 0.04 | 2983 | 664 | 14.00 | 2.19 | 12.35–23.43 |
| March | 26 | 6.57 | 1.94 | 3.73–11.58 | 0.07 | 0.02 | 4193 | 700 | 12.20 | 0.87 | 12.00–17.95 |
| April | 22 | 6.07 | 1.79 | 3.45–10.70 | 0.02 | 0.004 | 6643 | 2356 | 12.05 | 0.75 | 12.00–16.87 |
| May | 26 | 24.43 | 7.20 | 13.87–43.03 | 0.01 | 0.003 | 3711 | 1471 | 12.03 | 0.74 | 12.00–16.48 |
| Total | 85 | 5.32 | 1.56 | 3.02–9.36 | 0.02 | 0.003 | 6474 | 1065 | 12.00 | 0.71 | 12.00–12.11 |

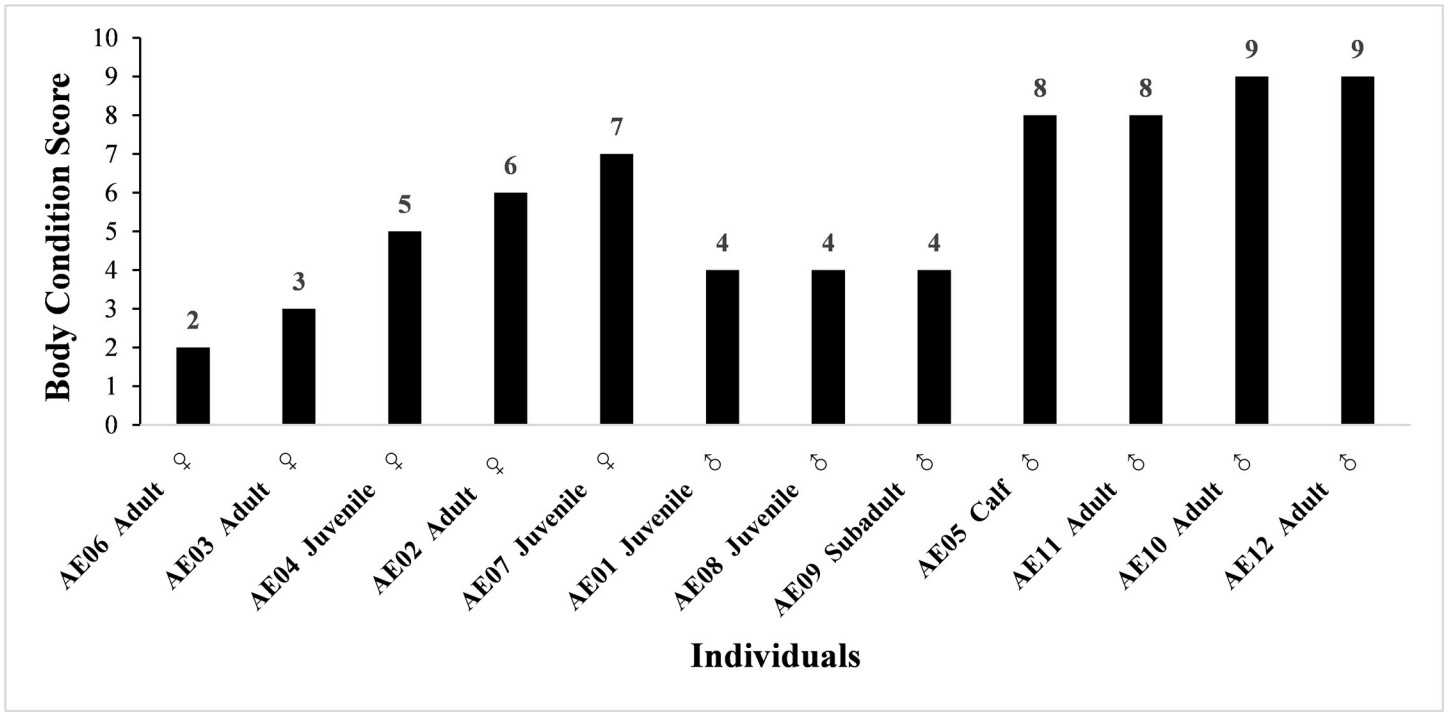

**Fig 3. Body condition score of the 12 individuals identified in NNNR.**

one tusker and two without tusks (known as makhna), giving an adult tusked to tuskless male ratio of 1:2. The subadult male was also a makhna, increasing the tusked to tuskless male ratio to 1:3. The ratio of reproductive ability was 1:0.67.

The social network was comprised by AE01-AE08 that formed a joint-family unit. Calf AE05, adult female AE06 and juvenile female AE07 formed a mother-calf unit which occasionally departed from the large family unit. AE06 was the mother of AE05, evidenced by images of the calf nursing from the female. The solitary subadult male AE09 and adult male AE10 occasionally joined the family unit.

## Body condition scoring

The average population BCS was 5.75 (n = 12, range 2–9; Fig 3). 33.3% of the elephants were in a good body condition, while 50% and 16.7% were in medium and poor body conditions, respectively (Fig 3). Age was not strong correlated to BCS ($r$ = 0.03), but sex was ($r$ = 0.42), as adult males had better body conditions than adult females. None of the only three fertile females in this population was classed as having a good BCS. AE02 had a BCS of 6 just above the average score, while AE03 and AE06 were in poor body conditions with a low BCS of 3. Individuals that were part of the herd had a lower BCS than solitary individuals ($r$ = 0.53).

## Discussion

### Population size and density

Accurate assessment of population size is essential for species conservation and management [65]. Identification from captured images was robust enough to identify individuals [66], unlike dung and footprints which are easily influenced by slope, soil quality and climatic factors [67]. This study used cameras installed in elephant preferred habitats that covered most of

**Table 5. Unnatural deaths of Asian elephant since the establishment of NNNR.**

| Year | Number of individuals | Cause of death | Data resources |
|------|----------------------|----------------|----------------|
| 1987 | Two: One female and one male | Illegal killing | [50, 71] |
| 1988 | Four: Two adult males, one adult female and one calf | Poaching, electrocuting and unknown reason | [47] |
| 1995 | One male | Unknown | [49] |
| 1996 | One male | Retaliatory killing by local people | [48] |
| 1997 | Three: One female, one male and a calf | Unknown | [50] |
| 2003 | A Subadult male (about 10 years old) | Unknown | [50] |
| 2008 | One subadult male | Unknown | Communication with local residents and rangers |
| 2011 | One juvenile | Unknown | [50] |
| 2016 | One adult male | Unknown, but the tusks showed visible saw marks | Communication with local residents and rangers |

the western part of the reserve. This provided a higher chance of obtaining data reflecting the real population, leading to a more reliable population assessment than in the study that used drone surveys [49]. Previous studies have estimated this population to be around 20 individuals. Although population estimation using different methods could generate bias in direct comparison, research has shown that this elephant population has remained small during the last few decades [49, 67–69] (S1 Table). The NNNR population is an extremely small and isolated population, compared with other elephant populations found around the world [3, 9, 10, 12]. Historic illegal hunting has had a direct impact on this population with a total of 15 unnatural deaths recorded since 1987 (Table 5). Additionally, 3 of 4 elephants found dead since 2003 have been immature individuals (Table 5). Moreover, the fact that elephants have a long adolescence and long inter-birth intervals means it takes time for elephants to recover and they are more vulnerable to local population extinction [12, 70], especially if young individuals are being poached or dying from other unnatural causes. All these facts result in the current small population size that could lead to the extinction of this local population.

SECR modelling has become a powerful tool to estimate the size of small populations like the one in NNNR [32]. The model defines the activity center according to the range of the species activities [72], so it is easy to overestimate the population density in a short survey period, which was confirmed by the monthly estimates in this study (Table 4). We first used single monthly sessions to run the SECR models, with which we obtained a strong difference in density estimation. The SECR models simulated a larger movement buffer (σ = 6643 m) in April, which resulted in the smallest population density. May presented the highest density because 8 cameras captured elephants, and the distance between each one was less than 800 m, which is smaller than their daily movements [73, 74]. The fluctuation in estimates from single month session results by SECR prompted us to use the combined four months' data as the fifth session to run the model. Parameter estimates from this session were the most accurate of the five [39].

We compared these results with those from other populations in order to improve our understanding of elephant ecology and develop more detailed management and conservation plans [75]. However, with no other SECR analyses conducted on other populations in China, we used the crude density estimation (estimated population number/study area) to draw comparison between the NNNR population and others in Yunnan [39]. The crude density estimation of Asian elephants in NNNR was the highest at 16.9 elephants/100 km$^2$, compared with that in Shangyong (69 individuals, 646 km$^2$), Mengyang (82 individuals, 998.4 km$^2$) and Mengla (40 individuals, 939.9 km$^2$) with a density of 10.68, 8.21, and 4.26 elephants/100 km$^2$, respectively [13, 17]. The relatively high population density might lead to more intensive competition among individuals for limited natural resources, and even greater human-elephant

conflict as elephant resort to crop raiding for food, which could be further studied in the future.

## Population structure in NNNR

The juveniles in this population accounted for 33% of the population, higher than the minimum of 20% needed for the population to be self-sustaining over a short period [18]. However, a previous study confirmed these individuals were related to each other [76]. Inbreeding will be more serious in the near future due to the small population size.

We could not match adult females with their offspring in the groups relying on observed behaviors alone, even though we captured intimate interaction between AE02 and AE08, as well as between AE03 and AE04. In recent years, the death of adult females has not been recorded in the reserve (Table 5). Therefore, we assumed that all the current juveniles were the offspring of these three adult females. Genetic studies are urgently needed to better understand the lineage and degree of inbreeding depression in the NNNR population.

Two tuskless juveniles were observed with body heights similar to that of the tusked juvenile males. These were identified as females, though it should be noted that tuskless males can occur in populations subjected to intensive historical poaching pressure [12, 77]. Since makhnas are common in this population, it would be helpful to conduct genetic studies on this population that can confirm the sex of these two juveniles.

The unbalanced sex ratio skewed towards males is not beneficial for population growth [78]. Male elephant can recognize kin and avoid inbreeding, and so male-biased populations usually indicate an increased genetic diversity and an ability to persist [79]. However, without introductions from elsewhere, such a small population with only 7 males is likely to go extinct in the near future. Females do not enter the breeding period until they are 10–14 years old and as such the two juvenile females in the NNNR population still need 3–7 years to become mature [9, 12, 80]. Currently, only females AE02, AE03 and AE06 are of reproductive age, but the latter was nursing a calf, leaving only two sexually active females in the reserve. In either case, the scarcity of fertile females would be the primary constraint on population growth due to their long birth intervals.

## Health condition

We chose a simple but accurate body condition method to assess the health of this small population. Our results showed a higher body condition of adult males than adult females. This may be because adult males are known to have larger home ranges than adult females [74, 81], and are more tolerant to human disturbance [82]. This means they can more frequently forage in areas near the reserve boundary, where intensive anthropogenic activities like hunting, grazing, fishing and collecting mushrooms have also been recorded by cameras. Additional nutritional stress from lactation, as observed in AE06, can also cause a poorer body condition than in males [44]. Furthermore, the presence of a calf can limit movement for members of the female members of the herd [81], while the solitary males have no such limitations. Additionally, the other two adult females both presented a medium health condition, which would not be conducive for reproduction [42].

Eight individuals (66.7%) had medium or poor body conditions during the survey period, which corresponded to the dry season. Since the quantity and quality of food and availability of water are the primary factors determining the body condition of free-ranging elephants [42], both female and male individuals show an obviously thinner body condition during the dry season compared to the rainy season [83]. We did not have the opportunity to conduct camera trap surveys during the rainy season since many cameras failed to work in this season.

Additionally, the rainy season coincides with the harvest period. In the past, especially in recent years, NNNR elephant-damage compensation data has shown that the elephants forage for corn and rice during the rainy season from July to October (Li Zhimin, personal communication). Therefore, we assume that at least some males in this population are foraging crops to supplement their diet. Future studies should focus on determining how crops contribute to improving the body condition of elephants during the rainy season.

Body condition assessment in this study could have implications for the research and conservation of other elephant populations. Since the vast majority of Asian elephant populations, especially in China, are found outside of reserves and near cropland, the health of these populations is likely heavily influenced by access to crops [84]. Massive amounts of data have been collected by camera traps and drones in these areas [85, 86], but none have been used to determine the health of elephant populations. We have shown that using images to assess BCS is a simple and fast method for determining the health of free-ranging elephant populations. Therefore, future studies should consider analyzing the body condition of regional Asian elephant populations from these data sets to obtain early indications of the health of individuals and quality of food resources in their habitats [43]. This in turn can be used to guide conservationists and policymakers to evaluate the necessity for interventions, including the restoration of suitable habitat [42].

## Conservation recommendations

The long-term placement of cameras in NNNR could be an efficient approach of monitoring to obtain valuable data on demography, fecundity and body condition variation across seasons and years. Our approaches can also be applied to other elephant populations with further implications for global Asian elephant conservation. Molecular biological approaches are also needed in further studies to determine the kinship and genetic health status of this population.

The results of this study match those observed in earlier studies which indicated providing more suitable habitat is urgently needed [13, 47, 48], this is especially imperative if the lower BCS scores are due to an inadequate supply of food. Attention should be paid to the low body condition in the dry season, and a follow-up surveys should be carried out in the rainy season to verify the health status of this elephant population throughout the year. To improve elephant survival in the short term, we recommend establishing more food sources and improving the quality of existing sources to increase individuals' body condition [87]. In the mid-term, fence dismantlement should be prioritized because it has partially blocked the movement of elephants and hindered their ability to find new food resources (Fig 1). Allowing the animals greater movement could provide more opportunities for the elephants to explore new suitable areas with enough food, eventually improving elephant body condition regardless of the season. Long-term objectives should include establishing ecological corridors to connect this isolated elephant population to other suitable habitat. Since the western part of this reserve (Fig 1. Red part of NNNR) has some suitable habitat [48], an ecological corridor could be created by planting bamboo forest or bamboo-broadleaf forests, allowing individuals to move between the eastern and western parts of the reserve. We also suggest conducting research on the availability of food resources during different seasons. We emphasize the urgency of implementing these recommendations, especially efforts to ensure a better body condition of females, which may improve reproductive success in the long run.

## Supporting information

**S1 Table. Historical elephant population in NNNR estimated by different methods.**
(DOCX)

**S1 Dataset. A simplified database of Asian elephant in NNNR.**
(PDF)

## Acknowledgments

This study was implemented by Beijing Forestry University and Beijing Eco-Bridge Continental. Furthermore, the field work was strongly supported by the administrators of Nangunhe National Nature Reserve, for which we thank them. We also thank all the field staff, including Jiahui Wang, Zhen Liu, Ying Liu and all local rangers, especially Deming Chen, Yongxiang Li, Shaobing Yang, Zhisheng Wang, Chunlian Li and Zhimin Li, for their hard and invaluable work.

## Author Contributions

**Conceptualization:** Ying Chen, Kun Shi.

**Data curation:** Kun Shi.

**Formal analysis:** Yakuan Sun.

**Funding acquisition:** Kun Shi.

**Investigation:** Yakuan Sun.

**Methodology:** Yakuan Sun, Ying Chen, Juan José Díaz-Sacco.

**Project administration:** Kun Shi.

**Resources:** Yakuan Sun.

**Software:** Yakuan Sun.

**Supervision:** Juan José Díaz-Sacco, Kun Shi.

**Validation:** Ying Chen.

**Visualization:** Yakuan Sun.

**Writing – original draft:** Yakuan Sun.

**Writing – review & editing:** Yakuan Sun, Ying Chen, Juan José Díaz-Sacco, Kun Shi.

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
