## [Decision Letter · Decision Letter 0]

19 Jun 2020

PONE-D-20-11590

Population structure and body condition assessment to inform conservation strategies for a small isolated Asian elephant population in southwest China

PLOS ONE

Dear Dr. Shi,

Thank you for submitting your manuscript to PLOS ONE. After careful consideration, we feel that it has merit but does not fully meet PLOS ONE’s publication criteria as it currently stands. Therefore, we invite you to submit a revised version of the manuscript that addresses the points raised during the review process.

Please, pay attention to all the comments from reviewers, both of them have done an exhaustive and positive work. Pay especial attention to comments of reviewer #2. This reviewer ask for improvement of grammar to enhance the information delivering. The methodology has missing information. Third, the discussion has too many speculative elements that are not supported by the results. Furthermore, the argumentation in the discussion is not always clear, which leaves the reader unsatisfied or unconvinced. 

We look forward to receiving your revised manuscript.

Kind regards,

Paulo Corti, Ph.D.

Academic Editor

PLOS ONE

Journal Requirements:

3. We note that Figure 1 in your submission contain map images which may be copyrighted. All PLOS content is published under the Creative Commons Attribution License (CC BY 4.0), which means that the manuscript, images, and Supporting Information files will be freely available online, and any third party is permitted to access, download, copy, distribute, and use these materials in any way, even commercially, with proper attribution. For these reasons, we cannot publish previously copyrighted maps or satellite images created using proprietary data, such as Google software (Google Maps, Street View, and Earth). For more information, see our copyright guidelines: http://journals.plos.org/plosone/s/licenses-and-copyright.

You may seek permission from the original copyright holder of Figure 1 to publish the content specifically under the CC BY 4.0 license. 

If you are unable to obtain permission from the original copyright holder to publish these figures under the CC BY 4.0 license or if the copyright holder’s requirements are incompatible with the CC BY 4.0 license, please either i) remove the figure or ii) supply a replacement figure that complies with the CC BY 4.0 license. Please check copyright information on all replacement figures and update the figure caption with source information. If applicable, please specify in the figure caption text when a figure is similar but not identical to the original image and is therefore for illustrative purposes only.

Reviewers' comments:

Reviewer's Responses to Questions

**Comments to the Author**

1. Is the manuscript technically sound, and do the data support the conclusions?

Reviewer #1: Yes

Reviewer #2: Partly

2. Has the statistical analysis been performed appropriately and rigorously? 

Reviewer #1: I Don't Know

Reviewer #2: Yes

3. Have the authors made all data underlying the findings in their manuscript fully available?

Reviewer #1: Yes

Reviewer #2: Yes

4. Is the manuscript presented in an intelligible fashion and written in standard English?

Reviewer #1: Yes

Reviewer #2: No

5. Review Comments to the Author

Reviewer #1: GENERAL

Overall a well-written and interesting study. Though I am unfamiliar with the SECR methodology, it is described in good detail. There are, however, a few extra details that the methods require for clarity, particularly in relation to the body condition scores (BCS). Results set out the basic demographics of this isolated population, and the conservation recommendations are clear.

SPECIFIC COMMENTS

- On line 220, the buffer zone is said to be 5km around the outermost camera sites. There was no citation here, so what was the justification for using 5km and not a different value?

- Lines 245-246 Please list all the model covariates and provide some description e.g. was age a categorical factor in the models? Was BCS considered as a continuous variable for the purposes of getting the correlation coefficients?

- It is unclear how many photographs were used to calculate the scores for each elephant. If multiple, were they all from around the same time and consistent? If the score is an average or not needs to be stated.

- Lines 285-287 Asian elephants may allolactate (Rapaport, L. & Haight, J. 1987 Some observations regarding allomaternal caretaking among captive Asian elephants (Elephas maximus). J. Mammal. 68, 438–442), and there may be other forms of allomothering too (Lahdenperä M., Mar, K.U., Lummaa V. 2016 Nearby grandmother enhances calf survival and reproduction in Asian elephants. Scientific Reports 6: 27213; Lynch E. C., Lummaa, V., Htut W., Lahdenperä M. 2019 Evolutionary significance of maternal kinship in a long-lived mammal. Phil. Trans. R. Soc B. 374: 20180067) – I’m not sure that any relationship can be considered ‘definite’

Reviewer #2: General comments

This study presents interesting information on a very tiny, endangered population of Asian elephants. The authors take advantage of movement-triggered cameras to get a wealth of information about this population, namely its size, structure, and a crude estimate of individual health. They get robust estimates of the populations thanks to the fact that the population seems completely isolated (and is small), and by using a state-of-the-art statistical method. However, the manuscript suffers from several problems that should and can be attended without too much difficulty. First, I must mention that some parts of the manuscript are difficult to read because of the way sentences are written. Sometimes it is clear that a single sentence should have been split in two to make sense, but in other opportunities, it was almost impossible to know what was meant. I suggest the authors carefully review the manuscript and ask a fluent English speaker knowledgeable in the field to revise the English (disclaimer: English is not my first language and I always ask a native speaker to revise, there is no shame in that). Second, the methodology has missing information or is not very clear in many parts. Third, the discussion has too many speculative elements that are not supported by the results. Furthermore, the argumentation in the discussion is not always clear, which leaves the reader unsatisfied or unconvinced. I believe that by restructuring the discussion and reducing it to the essential information supported by your results would improve its clarity. In conclusion, in my opinion this is a piece that deserves publication by providing key information on a small endangered population based on a technique now readily available to most organisations in charge of conservation. As such it exemplifies how this tool can be used to inform managers.

Specific comments

Here I list the comments I have done in the manuscript. I have also made a large number of suggestions of corrections directly in the manuscript, which I sent to the Associate Editor as they would not appear in the pdf version produced by the system in Editorial Manager.

L. 34: Insert BCS here for the first time.

L. 36: The fact that individuals had such low BCS should be highlighted in the abstract (if space allows).

L. 44: End the sentence at “dispersal” and start a new sentence about elephants as umbrella species.

L. 49: When you mention that the species is listed as Endangered in IUCN Red List, and then that its population is decreasing, you make a tautological argument. Any endangered population is decreasing. You should rather change the sentence and say that however important it is for conservation (as an umbrella species and an ecosystem engineer), there is a problem with the species, which population is decreasing, hence justifying its endangered status.

L. 77: Puma is not a good example, as they do not have spots or stripes. I would also keep this list more general, citing genera that have spots or stripes, such as tigers, leopards, giraffes, etc.

L. 88: There is a jump between the two sentences (no logical sequence). Maybe you could close the previous argument by saying that for NNNR there are not such data available. Then you could start a new paragraph on this isolated population, its genetic status, etc.

L. 99-108: This new paragraph also makes a jump. You should place it in the series of paragraphs describing how trap cameras (and photographs more broadly) are another way that tool can be useful in conservation and management.

L. 104-105: It is unclear why BCS is most meaningful “especially [in] small isolated populations”. Explain it or delete it.

L. 107-108. Delete this sentence. It is of very little relevance for a journal with a broad, worldwide audience.

L. 109-116: There is one more reason to move the previous paragraph upward: this paragraph is connected to the one preceding the previous one (i.e., after the paragraph l. 82-98).

L. 114-115: “for the development of a successful project focusing on the introduction of individual elephants from other populations”. That is the kind of sentence that does not serve you well. You talk about introducing individuals, but you have provided no evidence that the conditions to ensure the survival of introduced animals are met in the area. In many areas, habitat (quantity or quality) or negative relationships with humans do not ensure survival, making the prospect of reintroduction a loosing strategy.

L. 120: I would change “to analyze the factors” to reflect that it is indeed a discussion of some factors that might have affected the population that you will present.

L. 122: Why in China only and not in other parts of the species distribution?

L. 134: Give properly the geographical coordinates (including N to the latitude, and W to the longitude).

L. 137: Please give a proper annual precipitation and temperature (a single value), or explain the range you provide.

L. 138: Why natural in “natural tropical forest”?

L. 140: By evergreen vegetation do you mean evergreen broadleaf species, in opposition to coniferous species?

L. 148: Use a better title, such as survey design or camera trap set-up.

L. 151: “along the animal trails with forest gaps where there were signs of elephants”. Unclear. Did you set up the cameras in forest gaps that were crossed by elephant trails? If so replace “with” by “in”. In any case, review this.

L. 152: “as well as other activities”. You cannot remain vague here; mention them all.

L. 152: Explain why you avoided certain areas. Was it because elephants do not use them, or because you had concerns for the safety of people in the field?

L. 149-164: In this section, there is information missing. As far as I understand you had one camera per site, but one sentence mentions that "at each site, camera traps were operated...", which suggests there were several cameras at each site. Also, there is information missing regarding the number of sites or cameras, the minimum distance that could separate two sites (because there could be spatial autocorrelation in your data). Finally, you should mention the camera model(s) you used in your study. Elephants are not small nor super fast, so trigger time should not be an issue, but still, getting information on the material used is necessary for replication purposes.

L. 166-169: The whole sentence needs to be rewritten for better clarity. After reading it, I am left with doubt about what constituted an independent event. I think I understood that an independent event could include several photographs of various individuals, with at least 5 h between such photographic sequences to make it temporally independent, but it is far from clear.

L. 177: Did you determine the sex in juveniles and calves? If so, how did you do that?

L. 185-187: How do you know they did not change? Ear can be teared at any moment; a tusk could be broken digging up... maybe it is the way it is written that makes it not very convincing. If you have an elephant with a particular tear in the ear, it is easy to "recapture" it, but if there was no marking, I am not sure how this could be reliable. I think a better explanation would help.

Table 1:

1) Regarding sexing, as mentioned earlier I think male and female should be labeled properly adult female and adult male, as the diagnostic characteristics are found in adult (and possibly subadult) individuals only.

2) “Sunken prominence”. How can a prominence be sunken?

3) Your classes for tusk length are ok, but I am not ok with the fact that you do not explain how you classified them in each of these categories. I am sure you had a criteria to decide (such as tusk length relative to the trunk).

L. 197: “The movement pattern of individuals activity center in study area”. This is unclear. Do you mean “The movement pattern of individual activity centers in the study area”, i.e., the movement pattern of the activity centers of a given individual evaluated at different times (sessions)? Sorry, but the devil is in the details!

L. 198-200: "the probability of detecting an individual at a particular detector (…) to the distance of the detector" makes no sense. Please rewrite.

L. 206 and elsewhere in this section: Use trap or detector, or explain that detectors are traps, but remain consistent throughout your explanations.

L. 217: Again, there is a problem with the phrasing and it is unclear what you mean.

L. 219-221: The whole sentence needs to be rephrased so that the reader can understand exactly what you did. Also “from the outermost x and y coordinates of the camera sites” is very unclear. A camera site is not a discrete point in space???

L. 224: Again, this is unclear. Is the file named "Capture histories"? Please check the sentence.

L. 225: Is it "occasion" (you haven't defined it previously) or "event"? Also, earlier you defined individual events as being separated by at least 5 hours. This means that several individual events could occur on a same day. How can your database account for that if your time unit is "day"?

L. 227: “whole data as five different sessions”. Again, it is unclear. Do you mean you divided your 4-month sampling period in 5 (i.e. each session would last 24 days)?

L. 236-244: The first of these five sentences is very unclear. Further, if you followed the method developed by 49 or 50, just cite them, and mention that the part on which the scoring focuses are the ribs, shoulders, etc. and say why. Usually it is because these parts can accumulate fat.

L. 244: Before mentioning which software you used to perform your analyses, say what kind of analysis you performed.

L. 244-245: Which type of correlation coefficient?

L. 245: “ratios of various covariates”: why ratios? Age is not a ratio, nor is sex or the solitary vs group status.

L. 246: “whether an individual was present or not in the herd”. Here you lost me... do you mean you checked the correlation between BCS and whether an individual was solitary or in herd? (note that this is not what you have written; you have written “the herd”)

Table 2:

1) Revise the title of this table.

2) Characters is incorrect here and in several places in the manuscript. The correct word here is characteristics.

L. 250-253: This is much better suited in Methods. I kept wondering how you set up your cameras, how many, and here is the answer. Yours results should start with what you obtained, not what you did.

L. 254: You never spoke of video before! Fix that in Methods or delete here if they were not used.

L. 261-264: You should revise that sentence. As it is, there is not much difference between the individual that was seen the least and the others. Rather give the mean and the range.

Fig. 2: This title does not correspond to what you describe in the text (i.e., the 12 identified individuals). I think it should say something like "Asian elephants individually identified in NNNR park, China”. Also, note that the way you cited the figure, one would expect to see the 12 individuals identified. Here you show only half of them.

Table 3:

1) Put “No” in the column for Tusk for each female. This is all the more important that you later discuss the fact that you might have misclassified some individuals.

2) Put “Herd” in each file for individuals AE01 to AE08.

L. 272-279: This is interesting, but falls out of the scope of the section (Individual identification).

L. 283-284: This sentence is not built correctly, and it is unclear what the 1:1 ratio stands for (is it a sex ratio too?). Also, as I mentioned previously, the way you mentioned the criteria for sexing seemed to refer to sexually mature individuals only, so I wonder how you determined sex in juvenile individuals, if the 1:1 ratio is indeed a sex ratio. This needs to be clarified in the methods too!

L. 285: What are “containment relationship family units”?

Table 4: In the title you mention five sessions, but in the table there are only four sessions, one for each sampled month.

L. 306-307: This is the definition of the median!!!!! Please, take this out, or give the mean and here you could assess how skewed the distribution is.

L. 311-312: Say which sex had the highest BCI. Rephrase “solitary or not” (being in a herd or solitary), and explain which status was associated with having a lower or higher BCS.

Figure 3: The figure does not help much, and a table would probably be more efficient here.

Table 5: For the adult individual that died in 2003, how can you know “it died for its ivory”? Was it killed (with bullets inside the body)? You could find an individual with the tusks removed post-mortem.

L. 337-341: This is definitely not of interest for the readership of PLoS One.

L. 343: Use a reference to justify the link you make between the population structure and its future.

L. 343-345: Rephrase that sentence (self contradiction the way it is written), and note that this sentence contradicts the previous one (speaks of a possibly stable population, when earlier you say that it indicates a decline).

L. 345-346: This sentence comes out of the blue. Link it to an argument. You can actually use it to finish the following sentence with this argument of inbreeding depression in small populations.

L. 351-352: Again, a sentence out of the blue, which has no direct link with your results. What argument do you want to discuss in this paragraph? Start with that, and then develop the argument.

L. 355: Why do you say even if it was 1:1.5?

L. 359: Ref. 72 is not about balance in sex ratio in Asian elephant, it is about a sexing method. Make sure it is appropriate to cite it.

L. 360-361: In this sentence you refer to tuskless males, but who identified these tuskless males? If it is another study, cite it.

L. 366: “116 individual deaths (…) might account for tuskless males”. It is unclear here what you mean.

L. 360-370: I find all this difficult to follow because of some problems of clarity, but above all, you discuss a point when you recognize that you do not know for sure if the tuskless individuals you identified were males or females!

L. 379: Add at the end of that sentence that this is especially true due to birth intervals.

L. 398-399: OK, but the way you present it by comparing to India leaves the reader with the idea that these authors (ref 79) did the study over a short period of time. Was this the case, if not, why discuss this and in that way? This is very troubling.

L. 401: This value (800 m distance) makes sense only if we have an idea of elephant activity areas and movements.

L. 404: Not real, but actual. It is still based on what you captured on photographs.

L. 405: estimated by whom? Because your sentences are not well linked, it is hard to follow.

L. 408: About carrying capacity: well, you said before that 40 years ago the estimate was of 20 individuals, so at 12 it is obvious that you are below (even if we don't know if 20 was at carrying capacity or not).

L. 394-410: All this paragraph is convoluted. Why make weird assumptions (two sentences above) if you know that the population has already been that size.

L. 415-418: Rewrite these two sentences, and put what is a result in the corresponding section.

L. 422-424: So what? Here it is not the males venturing into human areas, but rather humans venturing into elephant territories. It means that males are more tolerant to humans, and has nothing to do with the quest for better food as when they venture in crop areas. You need to thread your arguments better.

L. 424-425: What you mention is well known, but you said that the other females were not lactating, so how does your argument hold?

L. 426: What does “reasonable” mean here? Also, you cannot consider the median or the mean as a standard. You may have a very healthy or very unhealthy population and the mean (or the median) can vary quite a lot.

L. 427-429: Yes, this is true, but this would skew evaluation towards lower BCS for adults, compared to juveniles.... and you found the opposite.

L. 431-438: All this is very speculative, and not very convincing. If the animals raid cultures during the rain season it might be for different reasons, and if they don't during the dry season maybe is it because they find enough food within the reserve. You have to be very careful: you do not have enough information to really discuss all that without speculating.

L. 438-441: Yes, but you have nothing on that: your data is from one season, in one particular year... not much to inform conservation managers.

L. 445-446: Take out the conflict part. It is not part of this argument and just distract the reader.

L. 448: “obesity level”. Surely you want to say something else! In nature animals are not obese (or very rarely), only in zoos.

L. 442-451: All this is well beyond what your data can tell.

L. 454: You keep mentioning the question of evenness in the sex ratio, but there is no foundation in your manuscript to justify why an even structure would be better.

L. 472-474: About reintroduction. Your study does not confirm that reintroduction is the solution for this population!!! In no way. See my previous comment on this in the introduction.

L. 478: Why mention restoration here? You have never mentioned that the habitat was either destroyed or degraded within the reserve. If it is the case, you should have better described your study area.

L. 485-486: About reintroduction to reduce inbreeding. You have no data on that, do not speculate, especially when recommending such drastic approach as translocation, which comes with lots of cons.

L. 489: Please restrain from mentioning reintroduction. Just managing the population to ensure its viability (if it is possible), would be enough! As I mentioned in the comments to the introduction, reintroduction is rarely a good solution, as the problem is usually with the habitat (quality or quantity) or with the negative interactions with humans.

Figure 3:

1) Eliminate “points” (after Presence, Absence, and Village).

2) The border with Myanmar and the limits of the reserve are similar (same width, same colour).

3) It is har to say where the experimental zone is on the map. It would be better to use the light shade to shade the area.

---

## [Author Response · Author response to Decision Letter 0]

7 Oct 2020

Specific Comments of Reviewer #1:

Comment 1: On line 220, the buffer zone is said to be 5km around the outermost camera sites. There was no citation here, so what was the justification for using 5km and not a different value?

Response: Considering the small size of the elephant distribution, with the east and west boundary being no wider than 10 km, we therefore defined our study area as extending 5 km beyond all camera stations, corresponding to an area of 255.8 km2. We cited one reference. Please go to P12, L226-228.

Comment 2: Lines 245-246 Please list all the model covariates and provide some description e.g. was age a categorical factor in the models? Was BCS considered as a continuous variable for the purposes of getting the correlation coefficients?

Response: The covariates include sex, age and whether the elephant was solitary or in the herd. BCS was not a continuous variable. Please go to P13, L260-262.

Comment 3: It is unclear how many photographs were used to calculate the scores for each elephant. If multiple, were they all from around the same time and consistent? If the score is an average or not needs to be stated.

Response: Five of the clearest images for each physical characteristic were selected to assess the body condition. Thirty images were used to calculate the comprehensive scores for each individual. Please go to P13, L244-246.

Comment 4: Lines 285-287 Asian elephants may allolactate (Rapaport, L. & Haight, J. 1987 Some observations regarding allomaternal caretaking among captive Asian elephants (Elephas maximus). J. Mammal. 68, 438–442), and there may be other forms of allomothering too (Lahdenperä M., Mar, K.U., Lummaa V. 2016 Nearby grandmother enhances calf survival and reproduction in Asian elephants. Scientific Reports 6: 27213; Lynch E. C., Lummaa, V., Htut W., Lahdenperä M. 2019 Evolutionary significance of maternal kinship in a long-lived mammal. Phil. Trans. R. Soc B. 374: 20180067) – I’m not sure that any relationship can be considered ‘definite’.

Response: Yes, we lost sight of that. Only the DNA methods can tell us the true relationship among individuals in this population. The word “definitely” has been deleted. We discussed this situation in the Discussion section. P16, L297-301; P20, L370-379.

Specific Comments of Reviewer #2:

Comment 1: L. 34: Insert BCS here for the first time.

Response: We have inserted BCS. This sentence has been changed to “The health condition of this elephant population was assessed by using an 11-points scale of Body Condition Scoring (BCS) method which showed that this population had a medium body condition with an average of 5.75 (n=12, range 2-9).” Please go to P2, L33-35.

Comment 2: The fact that individuals had such low BCS should be highlighted in the abstract (if space allows).

Response: We have rewritten this sentence. Please see the response to comment 1 or P2, L33-35.

Comment 3: L. 44: End the sentence at “dispersal” and start a new sentence about elephants as umbrella species.

Response: This sentence has been changed as you recommended. Please go to P3, L42-46.

Comment 4: L.49: When you mention that the species is listed as Endangered in IUCN Red List, and then that its population is decreasing, you make a tautological argument. Any endangered population is decreasing. You should rather change the sentence and say that however important it is for conservation (as an umbrella species and an ecosystem engineer), there is a problem with the species, which population is decreasing, hence justifying its endangered status.

Response: Yes, we believe that this is more logical. This sentence has been changed to make it simpler. “It is a Class I Key Protected Species in China and listed as an Endangered under the IUCN Red List.”. Please go to P3, L48-49. 

Comment 5: L.77: Puma is not a good example, as they do not have spots or stripes. I would also keep this list more general, citing genera that have spots or stripes, such as tigers, leopards, giraffes, etc.

Response: Puma has been replaced by leopard cats since they have stripes on their bodies. Please go to P4, L76.

Comment 6: L.88: There is a jump between the two sentences (no logical sequence). Maybe you could close the previous argument by saying that for NNNR there are not such data available. Then you could start a new paragraph on this isolated population, its genetic status, etc.

Response: A new paragraph has been created. Please go to P5, L99.

Comment 7: L.99-108: This new paragraph also makes a jump. You should place it in the series of paragraphs describing how trap cameras (and photographs more broadly) are another way that tool can be useful in conservation and management.

Response: We moved the BCS paragraph forward placing it behind the paragraph that introduces individual identification by camera traps. Please go to P4-5, L81-92. 

Comment 8: L.104-105: It is unclear why BCS is most meaningful “especially [in] small isolated populations”. Explain it or delete it.

Response: We deleted the word “small”. Now the sentence looks like “While BCS is individual-based, it is most meaningful when applied to a population, as an early indicator of the impact of management actions on the average health condition of a population.”. Please go to P5, L87-90. 

Comment 9: L. 107-108. Delete this sentence. It is of very little relevance for a journal with a broad, worldwide audience.

Response: This sentence has been deleted. Please go to P5, L92.

Comment 10: L.109-116: There is one more reason to move the previous paragraph upward: this paragraph is connected to the one preceding the previous one (i.e., after the paragraph l. 82-98).

Response: That is correct. After moving the previous paragraph upward, this paragraph is now well connected to the previous and following paragraphs. Please go to P6, L109-115.

Comment 11: L. 114-115: “for the development of a successful project focusing on the introduction of individual elephants from other populations”. That is the kind of sentence that does not serve you well. You talk about introducing individuals, but you have provided no evidence that the conditions to ensure the survival of introduced animals are met in the area. In many areas, habitat (quantity or quality) or negative relationships with humans do not ensure survival, making the prospect of reintroduction a loosing strategy.

Response: We did not obtain enough evidence to support the introduction of individuals as a successful plan for the survival of this small population. Thus, we deleted any mention about the introduction of elephants throughout the manuscript. Please go to P6, L112-115 and Conservation Recommendation section.

Comment 12: L. 120: I would change “to analyze the factors” to reflect that it is indeed a discussion of some factors that might have affected the population that you will present.

Response: This sentence has been changed to “We sought to understand which factors might have affected this populations size in recent years in order to develop targeted and effective protection strategies.”. Please go to P6, L118-119.

Comment 13: L. 122: Why in China only and not in other parts of the species distribution?

Response: We have deleted “in China”. The new sentence is “Furthermore our approaches can be applied to other elephant populations with further implications for global Asian elephant conservation.” Please go to P6, 120-121.

Comment 14: L. 134: Give properly the geographical coordinates (including N to the latitude, and W to the longitude).

Response: The sentence had been changed to “It occupies 70.82 km2 at latitude 23°13'- 23°19'N, longitude 98°54'- 99°05'E, and the altitude ranges from 510 m to 1747 m.”. P7, L131-133. 

Comment 15: L. 137: Please give a proper annual precipitation and temperature (a single value), or explain the range you provide.

Response: Mean annual precipitation is 351.3 mm in the dry season and 1983.3 mm in the rainy season, while temperature varies at different altitudes. The average monthly temperature range is hottest in July, varying between 12.3-25.5°C and it is coldest in January with values between 2.8-14.3 ℃. (Reference: NGH scientific report. No single value for annual temperature, just range value.) Please go to P7, L134-137. 

Comment 16: L. 138: Why natural in “natural tropical forest”?

Response: It should be a tropical forest. We made a little mistake when we read the corresponding reference. Please go to P7, L138.

Comment 17: L.140: By evergreen vegetation do you mean evergreen broadleaf species, in opposition to coniferous species?

Response: Yes, it is evergreen broadleaf species. We have added “broadleaf” in the manuscript. Please go to P7, L141. 

Comment 18: L. 148: Use a better title, such as survey design or camera trap set-up.

Response: The title has been changed to “Camera trap set-up”. Please go to P7, L148. 

Comment 19: L. 151: “along the animal trails with forest gaps where there were signs of elephants”. Unclear. Did you set up the cameras in forest gaps that were crossed by elephant trails? If so replace “with” by “in”. In any case, review this.

Response: Actually, we set up the cameras in trails and forest gaps, but the gaps were often located in trails. So, here we replace “with” by “and” to make it clearer. Please go to P8, L155-157. 

Comment 20: L. 152: “as well as other activities”. You cannot remain vague here; mention them all.

Response: The elephant signs included lying sites, foraging sites, footprints, dungs and urine marks. Please go to P8, L155-157. 

Comment 21: L. 152: Explain why you avoided certain areas. Was it because elephants do not use them, or because you had concerns for the safety of people in the field?

Response: We avoided the northwestern part of the reserve as the topography is tough, with steep cliffs of up to 1747 meters, and historically, there have not been any records of elephants in this area. Please go to P8, L159-162. 

Comment 22: L. 149-164: In this section, there is information missing. As far as I understand you had one camera per site, but one sentence mentions that "at each site, camera traps were operated...", which suggests there were several cameras at each site. Also, there is information missing regarding the number of sites or cameras, the minimum distance that could separate two sites (because there could be spatial autocorrelation in your data). Finally, you should mention the camera model(s) you used in your study. Elephants are not small nor super fast, so trigger time should not be an issue, but still, getting information on the material used is necessary for replication purposes.

Response: A total of 52 cameras at 52 stations or sites (P8, L150-151). We set one camera trap at each camera site, but the number of camera sites varied in each grid. More than one camera was installed in the grids that contained lots of elephant signs (P8, L170-171), and the minimum distance between two cameras was 200 meters (P8, L158-159). The camera model was Ltl Acorn 6210 (P8, L150) and we programmed the camera to take sequential images consisting of three photographs plus one 12 second video (P8, L168-170). 

Comment 23: L. 166-169: The whole sentence needs to be rewritten for better clarity. After reading it, I am left with doubt about what constituted an independent event. I think I understood that an independent event could include several photographs of various individuals, with at least 5 h between such photographic sequences to make it temporally independent, but it is far from clear.

Response: We have provided a more detailed explanation about independent events than in the previous manuscript version. Please see P9, L175-181. 

Comment 24: L. 177: Did you determine the sex in juveniles and calves? If so, how did you do that?

Response: Yes, we made it clearer in Table 1. Please see P10, L199. 

Comment 25: L. 185-187: How do you know they did not change? Ear can be teared at any moment; a tusk could be broken digging up... maybe it is the way it is written that makes it not very convincing. If you have an elephant with a particular tear in the ear, it is easy to "recapture" it, but if there was no marking, I am not sure how this could be reliable. I think a better explanation would help.

Response: Yes, that is a better explanation that would be more convincing. Morphological characters including facial bones, ear, tusk, back and tail hair, old scars and injuries, as well as physical characteristics related to sex and age classes, confirmed the reliability of identification results. Please see P10, L196-198.

Comment 26: Table 1: 1) Regarding sexing, as mentioned earlier I think male and female should be labeled properly adult female and adult male, as the diagnostic characteristics are found in adult (and possibly subadult) individuals only. 2) “Sunken prominence”. How can a prominence be sunken? 3) Your classes for tusk length are ok, but I am not ok with the fact that you do not explain how you classified them in each of these categories. I am sure you had a criteria to decide (such as tusk length relative to the trunk).

Response: 1): We had rearranged sex categories as adult female, adult male, subadult male, juvenile female, juvenile male and calf male. 2): It was a description mistake. The sentence has been changed to “Extent of depression”. 3): Yes, we have a criterion that uses trunk as a reference to classify the tusk length. For more details please go to Table 1 on P10, L199.

Comment 27: L. 197: “The movement pattern of individuals activity center in study area”. This is unclear. Do you mean “The movement pattern of individual activity centers in the study area”, i.e., the movement pattern of the activity centers of a given individual evaluated at different times (sessions)? Sorry, but the devil is in the details!

Response: Yes, the model can evaluate the movement pattern according to the capture history. We missed the time limitation. The sentence has been changed to “The state model describes statistically the mechanisms determining the distribution of the animals in the study area, while the observation model quantifies the probabilities of detection or capture, given the searched region, their locations and characteristics at different sessions”. Please go to P11, L205-209. 

Comment 28: L. 198-200: "the probability of detecting an individual at a particular detector (…) to the distance of the detector" makes no sense. Please rewrite.

Response: The sentence has been rewritten combining it with the forward sentence. Please see the response of comment 27 or onP11, L205-209.

Comment 29: L. 206 and elsewhere in this section: Use trap or detector, or explain that detectors are traps, but remain consistent throughout your explanations.

Response: All the words corresponding to “detectors” were replaced by “traps”. Please go to P11, L212; L214; L217.

Comment 30: L. 217: Again, there is a problem with the phrasing and it is unclear what you mean.

Response: This sentence has been deleted. Please go to L12, L224. 

Comment 31: L. 219-221: The whole sentence needs to be rephrased so that the reader can understand exactly what you did. Also “from the outermost x and y coordinates of the camera sites” is very unclear. A camera site is not a discrete point in space???

Response: The sentence has been changed to “Considering the small size of the elephant distribution, with the east and west boundary being no wider than 10 km, we therefore defined our study area as extending 5 km beyond all camera stations, corresponding to an area of 255.8 km2.”. Please go to P12, L226-228.

Comment 32: L. 224: Again, this is unclear. Is the file named "Capture histories"? Please check the sentence.

Response: Yes, the file is named “capture histories”. The sentence has been changed. Please go to P12, L231.

Comment 33: L. 225: Is it "occasion" (you haven't defined it previously) or "event"? Also, earlier you defined individual events as being separated by at least 5 hours. This means that several individual events could occur on a same day. How can your data base account for that if your time unit is "day"?

Response: “Occasion” was defined as each event, as well as the “sequence”. The cameras captured several events, accounting for different occasions. Please go to P12, L232.

Comment 34: L. 227: “whole data as five different sessions”. Again, it is unclear. Do you mean you divided your 4-month sampling period in 5 (i.e. each session would last 24 days)?

Response: Five different sessions, corresponding to each of the months and one corresponding to the whole survey period, were used to run the model to see the estimation accuracy under different detection efforts. P12, L233-236.

Comment 35: L. 236-244: The first of these five sentences is very unclear. Further, if you followed the method developed by 49 or 50, just cite them, and mention that the part on which the scoring focuses are the ribs, shoulders, etc. and say why. Usually it is because these parts can accumulate fat.

Response: We added more information to describe the reasons we choose ribs, shoulders and other body parts as scoring criterion. Please go to P13, L246-252. 

Comment 36: L. 244: Before mentioning which software you used to perform your analyses, say what kind of analysis you performed.

Response: This sentence has been deleted. Please go to P13, L260.

Comment 37: L. 244-245: Which type of correlation coefficient?

Response: It is Pearson Correlation Coefficient. P13, L260.

Comment 38: L. 245: “ratios of various covariates”: why ratios? Age is not a ratio, nor is sex or the solitary vs group status.

Response: The word “ratio” has been deleted. Please go to P13, L260-262.

Comment 39: L. 246: “whether an individual was present or not in the herd”. Here you lost me... do you mean you checked the correlation between BCS and whether an individual was solitary or in herd? (note that this is not what you have written; you have written “the herd”)

Response: Yes, we wanted to divide individual elephants into those that were solitary and those that were part of a herd to see if this factor could affect the BCS. Please go to P13, L260-262. 

Comment 40: Table 2: 1) Revise the title of this table. 2) Characters is incorrect here and in several places in the manuscript. The correct word here is characteristics.

Response: 1): The title has been changed to “Table 2. Characteristics used to assess the body condition in Asian elephant, NNNR”. 2): All the words “characters” have been changed to “characteristics”. Please go to P13, L263 to see Table 2.

Comment 41: L. 250-253: This is much better suited in Methods. I kept wondering how you set up your cameras, how many, and here is the answer. Yours results should start with what you obtained, not what you did.

Response: Yes, these sentences presented in the Methods section were more logical. These sentences have been moved to their current location. P7-8, L149-152.

Comment 42: L. 254: You never spoke of video before! Fix that in Methods or delete here if they were not used.

Response: We apologize for this mistake. We programmed the camera to take three pictures and a 12 second video. We replaced the words “picture” and “video” by “image” in the Methods section. Please go to P8, L168-170. 

Comment 43: L. 261-264: You should revise that sentence. As it is, there is not much difference between the individual that was seen the least and the others. Rather give the mean and the range.

Response: We strongly agree with your suggestions. Currently, there is a brief comment about the capture histories of individual and herds. Please go to P14, L271-274. 

Comment 44: Fig. 2: This title does not correspond to what you describe in the text (i.e., the 12 identified individuals). I think it should say something like "Asian elephants individually identified in NNNR park, China”. Also, note that the way you cited the figure, one would expect to see the 12 individuals identified. Here you show only half of them.

Response: We strongly agree with your suggestions and the title of “Fig 2” was changed as you recommended. The figure was re-edited to combine all 12 individuals in one picture, and we used black squares to emphasize each individual’s prominent characteristic so as to make it clearer for readers. Please go to P15, L285-287 and the file named “Fig2”.

Comment 45: Table 3: 1) Put “No” in the column for Tusk for each female. This is all the more important that you later discuss the fact that you might have misclassified some individuals. 2) Put “Herd” in each file for individuals AE01 to AE08.

Response: The changes you proposed have been achieved in Table 3 of the new manuscript. Please go to P14, L275.

Comment 46: L. 272-279: This is interesting, but falls out of the scope of the section (Individual identification).

Response: Yes, we strongly agree with you and thus, this paragraph has been deleted after careful consideration.

Comment 47: L. 283-284: This sentence is not built correctly, and it is unclear what the 1:1 ratio stands for (is it a sex ratio too?). Also, as I mentioned previously, the way you mentioned the criteria for sexing seemed to refer to sexually mature individuals only, so I wonder how you determined sex in juvenile individuals, if the 1:1 ratio is indeed a sex ratio. This needs to be clarified in the methods too!

Response: We wrote this sentence incorrectly. The 1:1 ratio stands for juvenile individuals as we identified two juvenile males and two juvenile females. But we failed to make it clear in the Methods section. Please go to P16, L290-296.

Comment 48: L. 285: What are “containment relationship family units”?

Response: Since we wrote this sentence using the wrong words, it has been deleted. Please go to P16, L297.

Comment 49: Table 4: In the title you mention five sessions, but in the table there are only four sessions, one for each sampled month.

Response: We apologize for the way that it was written. First, we used single month as a single session, and then we used the whole months data as the fifth session. Please see more details in comment 34 or go to P12, L233-236.

Comment 50: L. 306-307: This is the definition of the median!!!!! Please, take this out, or give the mean and here you could assess how skewed the distribution is.

Response: This sentence has been deleted. Please go to P16, L304.

Comment 51: L. 311-312: Say which sex had the highest BCI. Rephrase “solitary or not” (being in a herd or solitary), and explain which status was associated with having a lower or higher BCS.

Response: This information has been combined. Please go to P16-17, L307-313. 

Comment 52: Figure 3: The figure does not help much, and a table would probably be more efficient here.

Response: We think a figure would be clearer and more visible than a table. We have improved the figure by changing the order of individuals by placing individuals of the same sex together and ranking them by scores. Please go to the file named “Fig3”.

Comment 53: Table 5: For the adult individual that died in 2003, how can you know “it died for its ivory”? Was it killed (with bullets inside the body)? You could find an individual with the tusks removed post-mortem.

Response: We read these two references again and deleted one reference which provided us with incomplete information about an adult male that had died for its ivory without any citation nor detailed explanation. So, the description of “the adult male died for its ivory” has been deleted. Please go to Table 5 on P18, L338.

Comment 54: L. 337-341: This is definitely not of interest for the readership of PLoS One.

Response: These sentences have been deleted. Please go to P19, L364.

Comment 55: L. 343: Use a reference to justify the link you make between the population structure and its future.

Response: Four references have been added. Please go to P17, L323-327. 

Comment 56: L. 343-345: Rephrase that sentence (self contradiction the way it is written), and note that this sentence contradicts the previous one (speaks of a possibly stable population, when earlier you say that it indicates a decline).

Response: This sentence has been rephrased. Please go to P20, L366-369. 

Comment 57: L. 345-346: This sentence comes out of the blue. Link it to an argument. You can actually use it to finish the following sentence with this argument of inbreeding depression in small populations.

Response: This sentence has been combined. Please go to comment 56 or P20, L366-369.

Comment 58: L. 351-352: Again, a sentence out of the blue, which has no direct link with your results. What argument do you want to discuss in this paragraph? Start with that, and then develop the argument.

Response: We strongly agree with your comments. This sentence has been changed to “There were two tuskless juveniles that had similar body height as the juvenile males identified by their apparent tusks and these were identified as females. However, it should be noted that males in some populations tend to be tuskless for survival due to the intensive historical poaching pressure. The high proportion of makhnas in NNNR identified from this investigation could indicate that these juveniles that were identified as females might in fact be males.” Please go to P20, L380-385. 

Comment 59: L. 355: Why do you say even if it was 1:1.5?

Response: This mistake has been corrected. Please go to P16, L290-296. 

Comment 60: L. 359: Ref. 72 is not about balance in sex ratio in Asian elephant, it is about a sexing method. Make sure it is appropriate to cite it.

Response: These sentences and reference 72 have all been deleted. Please go to P17, L323.

Comment 61: L. 360-361: In this sentence you refer to tuskless males, but who identified these tuskless males? If it is another study, cite it.

Response: We identified these tuskless males. Approaches were displayed in Table 1 section. Please go to Table 1.

Comment 62: L. 366: “116 individual deaths (…) might account for tuskless males”. It is unclear here what you mean.

Response: High poaching pressure could lead to the predominance of tuskless males. Since currently there is a high proportion of tuskless males in this population, we had assumed that the reason for this situation was historical poaching. But since we don’t have data to support this assumption, this sentence has been deleted. Please go to P20, L383-385.

Comment 63: L. 360-370: I find all this difficult to follow because of some problems of clarity, but above all, you discuss a point when you recognize that you do not know for sure if the tuskless individuals you identified were males or females!

Response: Yes. We judged these juveniles as females under the condition that they have similar body height as juvenile males who had apparent tusks. And at the same time, we discussed that since tuskless males are common in this population, it would be helpful to conduct genetic studies on this population that can confirm the sex of these two juveniles. Please go to P20, L380-387.

Comment 64: L. 379: Add at the end of that sentence that this is especially true due to birth intervals.

Response: The sentence has been changed to “In either case, the scarcity of fertile females would be the primary constraint on population growth due to their long birth intervals.” P21, L394-395. 

Comment 65: L. 398-399: OK, but the way you present it by comparing to India leaves the reader with the idea that these authors (ref 79) did the study over a short period of time. Was this the case, if not, why discuss this and in that way? This is very troubling.

Response: It has been deleted. Please go to P19, L344.

Comment 66: L. 401: This value (800 m distance) makes sense only if we have an idea of elephant activity areas and movements.

Response: Here, we cited two references that pointed out that this value was smaller than elephant daily movement distance. Please go to P19, L346-348.

Comment 67: L. 404: Not real, but actual. It is still based on what you captured on photographs.

Response: Yes, the result should be actual value. The changes have been made in the new manuscript. Please go to P19, L350-351.

Comment 68: L. 405: estimated by whom? Because your sentences are not well linked, it is hard to follow.

Response: We estimated it. We found the reference by Sukumar (2003) who gave an assumption that a population density of 0.1-0.3 elephant/km2 was normal. So, we used this assumption to estimate there should be 20 individuals in NNNR (71 km2). In the new manuscript, we deleted this hypothesis. Please go to P17, L329-331.

Comment 69: L. 408: About carrying capacity: well, you said before that 40 years ago the estimate was of 20 individuals, so at 12 it is obvious that you are below (even if we don't know if 20 was at carrying capacity or not).

Response: The use of the word “carrying capacity” made the sentence less rigorous. This word has been deleted. Please go to P17, L327-331. 

Comment 70: L. 394-410: All this paragraph is convoluted. Why make weird assumptions (two sentences above) if you know that the population has already been that size.

Response: The whole paragraph has been rewritten. Thus, we combined the density section with population size section to make it simpler. Please go to the Population size and density part of the Discussion section. P18, L323-337.

Comment 71: L. 415-418: Rewrite these two sentences, and put what is a result in the corresponding section.

Response: These two sentences have been placed in the corresponding section. Please go to P16, L307-308. 

Comment 72: L. 422-424: So what? Here it is not the males venturing into human areas, but rather humans venturing into elephant territories. It means that males are more tolerant to humans, and has nothing to do with the quest for better food as when they venture in crop areas. You need to thread your arguments better.

Response: Thank you for your recommendation. This part has been rewritten. Please see P21, L400-401. 

Comment 73: L. 424-425: What you mention is well known, but you said that the other females were not lactating, so how does your argument hold?

Response: The only three adult females in this population were all considerd to be potentially productive, and the body health condition of all of them was considered to be average or even in poor health condition. Our data showed that the individual AE06 who was nursing an offspring had the lowest BCS in this population. Therefore, we mentioned additional nutritional stress from lactation for AE06 could have resulted in an even poorer body condition. Please go to P21, L404-409.

Comment 74: L. 426: What does “reasonable” mean here? Also, you cannot consider the median or the mean as a standard. You may have a very healthy or very unhealthy population and the mean (or the median) can vary quite a lot.

Response: This sentence has been deleted. We had classified the 11-point scale with three ranks: poor, medium and good. Please go to P13, L259-260. 

Comment 75: L. 427-429: Yes, this is true, but this would skew evaluation towards lower BCS for adults, compared to juveniles.... and you found the opposite.

Response: Yes, that is correct. Maybe the adults have lower BCS than juveniles assuming that we had a large population like previous studies. The truth is the population in NNNR was extremely small, and the fact is that we found that the adults had higher BCS than the juveniles. These sentences have been deleted since we do not want to confuse readers. Please go to P21, L406.

Comment 76: L. 431-438: All this is very speculative, and not very convincing. If the animals raid cultures during the rain season it might be for different reasons, and if they don't during the dry season maybe is it because they find enough food within the reserve. You have to be very careful: you do not have enough information to really discuss all that without speculating.

Response: We strongly agree with your comments. Please go to P22, L410-422.

Comment 77: L. 438-441: Yes, but you have nothing on that: your data is from one season, in one particular year... not much to inform conservation managers.

Response: This sentence has been deleted. Please go to P22, L419-422. 

Comment 78: L. 445-446: Take out the conflict part. It is not part of this argument and just distract the reader.

Response: This conflict part has been deleted. Please go to P22, L423-427.

Comment 79: L. 448: “obesity level”. Surely you want to say something else! In nature animals are not obese (or very rarely), only in zoos.

Response: We wanted to say “health condition”. Please go to P23, L432. 

Comment 80: L. 442-451: All this is well beyond what your data can tell.

Response: We aimed to extend our research to other elephant populations. Although more and more studies have been conducted in China, the study of the BCS has been neglected. Thus, we wanted to draw attention on this kind research for Chinese researchers and conservationists. Please go to P22-23, L423-435.

Comment 81: L. 454: You keep mentioning the question of evenness in the sex ratio, but there is no foundation in your manuscript to justify why an even structure would be better.

Response: We have deleted this part. Please go to P23, L436.

Comment 82: L. 472-474: About reintroduction. Your study does not confirm that reintroduction is the solution for this population!!! In no way. See my previous comment on this in the introduction.

Response: All the information related to reintroduction has been deleted. Please go to comment 11 and P6, L112-115 or Conservation Recommendation section.

Comment 83: L. 478: Why mention restoration here? You have never mentioned that the habitat was either destroyed or degraded within the reserve. If it is the case, you should have better described your study area.

Response: The word “restoration” has been deleted. Please go to Conservation Recommendation section on P436.

Comment 84: L. 485-486: About reintroduction to reduce inbreeding. You have no data on that, do not speculate, especially when recommending such drastic approach as translocation, which comes with lots of cons.

Response: All the information related to reintroduction has been deleted. Please go to comment 11 and P6, L112-115 or Conservation Recommendation section.

Comment 85: L. 489: Please restrain from mentioning reintroduction. Just managing the population to ensure its viability (if it is possible), would be enough! As I mentioned in the comments to the introduction, reintroduction is rarely a good solution, as the problem is usually with the habitat (quality or quantity) or with the negative interactions with humans.

Response: All the information related to reintroduction has been deleted. Please go to comment 11 and P6, L112-115 or Conservation Recommendation section.

Comment 86: Figure 3: 1) Eliminate “points” (after Presence, Absence, and Village). 2) The border with Myanmar and the limits of the reserve are similar (same width, same colour). 3) It is har to say where the experimental zone is on the map. It would be better to use the light shade to shade the area.

Response: All the suggestions that you proposed have been taken into consideration. See more details in file named “Fig 1”.

---

## [Decision Letter · Decision Letter 1]

30 Nov 2020

PONE-D-20-11590R1

Population structure and body condition assessment to inform conservation strategies for a small isolated Asian elephant population in southwest China

PLOS ONE

Dear Dr. Shi,

Thank you for submitting your manuscript to PLOS ONE. After careful consideration, we feel that it has merit but does not fully meet PLOS ONE’s publication criteria as it currently stands. Therefore, we invite you to submit a revised version of the manuscript that addresses the points raised during the review process.

Check suggestions from both reviewers below, but especially follow the comments of reviewer #2 and please solve them. I agree with those suggestions that I am sure they will improve your work.

We look forward to receiving your revised manuscript.

Kind regards,

Paulo Corti, Ph.D.

Academic Editor

PLOS ONE

Reviewer #1: In my opinion, this revised manuscript addresses all comments from myself and the other reviewer in a satisfactory way. I have two really small suggestions:

line 323 - 'as in' should be 'than in'

line 401-402 - You may want to include a short explanation for why larger home ranges and tolerance of disturbance could be linked to higher BCS. Additionally, the sentence starting 'Which means' should instead start 'Tolerance means'

Reviewer #2: 

General comments

The authors have thoroughly attended all the comments and corrections I suggested, and I commend them for this. The methods in particular are now very clear. The discussion, however, still suffers from several problems. First, some new sentences are very unclear; usually they are long, and the authors should be more straightforward. Again, it leaves the reader unsatisfied or unconvinced. As suggested in the previous revision, I strongly suggest the authors ask a fluent English speaker knowledgeable in the field to revise the English. Second, some paragraphs, especially in the first sections of the discussion, do not clearly develop an idea, so it is hard to understand their usefulness. Third, paragraphs themselves are not very well articulated, which make reading “bumpy”. The three recommendations, namely providing food sources during the dry season within the reserve (especially for females), dismantling the fence and establishing dispersal corridors between different areas of the park to allow dispersal to better feeding areas, are very interesting and should be the only focus of the last section. Platitudes such as continuing to document the populations using camera traps to save the population should be avoided, as it dilutes the interesting information.

Specific comments

All specific comments and corrections are included directly in the new version of the manuscript.

---

## [Author Response · Author response to Decision Letter 1]

20 Dec 2020

General Comments of Reviewer #1:

Response: Thanks for your recognition of our new version manuscript. Here were responses of your two small suggestions.

Comment 1: Line 323, 'as in' should be 'than in'

Response: The word has been replaced by the one your recommendation. Please go to P16, L313.

Comment 2: line 401-402 - You may want to include a short explanation for why larger home ranges and tolerance of disturbance could be linked to higher BCS. Additionally, the sentence starting 'Which means' should instead start 'Tolerance means'

Response: Large home ranges and tolerance of disturbance means the male elephants obtain more food resources, which contribute a good body condition in turn. The new sentence has been changed like “This may be because adult males are known to have larger home ranges than adult females, and are more tolerant to human disturbance: that means they can explore more frequently areas near the reserve boundary to obtain more food resources, where intensive anthropogenic activities like hunting, grazing, fishing and collecting have also been recorded by cameras.”

General Comments of Reviewer #2:

Response: Thank you for your exhaustive and positive comments and suggestions. We made huge changes on Discussion to make it clearer and logistical. In Conservation Recommendations part, we only used two sentences to describe the importance of long-term monitoring of infrared cameras. Then focus on the suggestions that improving food resources to enhance the elephant body condition. This paper has been checked by professional editors, both native speakers of English.

Comment 1: On line 93-95, This sentence should go Line 80, to finish the paragraph on photo ID.

Response: This sentence had moved to Line 80 in the new version of manuscript. Please go to P4, L80-82.

Comment 2: On the line 99-115, These two paragraphs should go right after the first one, when you describe elephant populations in China. Otherwise, these paragraphs do not connect well with the previous ones.

Response: We deleted some sentences in these two paragraphs and have rewritten remaining sentences. We thought these could be more logical and precise than before. Please go to P5, L95-105.

Comment 3：On the line 200, Maybe the way you present the table is not optimal, because when you describe the categories in SEX, you include already both sex and age. So maybe you should start describing AGE, and then SEX.

Response: We adapted your opinion by describing AGE first and then describing SEX which was simply categorized to female and male. Please go to P9-10, see Table 1.

Comment 4: On Table 1: Ok, so it is obvious that adult and subadult males that are tuskless may be impossible to distinguish; maybe add "and no penis", to distinguish them from their male counterparts; Same, you should add the testicles/penis part, but here, how do you distinguish juvenile from subadult males

Response: We have simplified the category of SEX by just describing female as “Without obvious tusks but with prominent breasts, belly swelling or the presence of nursing offspring” and male as “With visible testicles or penis, and/or obvious tusks”. See Table 1.

Comment 5: On the line 210, It should be replaced by "each", even when it is clear that you mean "each and every"

Response: It should be “each” in this situation. Please go to P11, L222. 

Comment 6：On the line 232, Not too clear to me. You mean to say "i.e."?

Response: This sentence has been changed into “The file named “capture histories” contained data on individual identified each sampling occasion from the specific detector (camera).” Please go to P11, L221-222.

Comment 7: On the line 323-327, This sentence is unclear. What do you want to convey here? Do you want to say that different methods lead to different results inveitably, or do you just want to point out that historically the size of the population was estimated at about 20 individuals?I think it is the second case, but in any case by straightforward.

Response: This sentence has been changed to make it simpler. “Previous studies have estimated this population to be around 20 individuals. Although population estimation using different methods could generate bias in direct comparison, research has shown that this elephant population has remained small during the last few decades.” Please go to P16-17, L313-317.

Comment 8: On the line 327-330, The logics here remains unclear; with the information provided it is not obvious why being in the same part of the park means that there should be more elephants. Maybe habitat quality is not that good anymore (in fact, the fact that females were not in good body condition might indicate something like that)... or it might be that the highest ever recorded population in the park was an artefact due to loss of habitat in the surrounding (and then a packing effect).

Response: We did not do any analysis in habitat quality estimation, so it is surprising to readers see the point suddenly. This sentence has been deleted.

Comment 9: On the line 335, You have four elephants dead since 2003... don't calculate statistics over that! Just say that 3 of the 4 elephants found dead since 2003 have been immature individuals.

Response: This sentence has been changed to “Additionally, 3 of 4 elephants found dead since 2003 have been immature individuals (Table5).”Please go to P17, L320-321.

Comment 10: On the line 335-336, This is part of the life history of the species, not somethinf particular to this population. The way it is written, it is misleading.

Response: This sentence has been rewritten to “Moreover, the fact that elephants have a long adolescence and long inter-birth intervals means it takes time for NGH elephants to recover and they are more vulnerable to local population extinction, especially if young individuals are being poached or dying from other unnatural causes. All these facts result in the current small population size that could lead to the extinction of this local population.” Please go to P17, L321-326.

Comment 11: On the line 362, Yes, which makes your suggestion that the population could be bigger even more bizarre.

Response: This sentence has been changed to “The relatively high population density might lead to more intense competition among individuals for limited natural resources, and even greater human-elephant conflict as elephants resort to crop-raiding for food, which could be further studied in the future.” Please go to P19, L350-353.

Comment 12: On the line 368-369, Why in the near future? Isn't it happening already? 

Response: This sentence has been changed to “Previous study already confirmed these individuals were related to each other [82]. This phenomenon of inbreeding will be more serious in the near future due to the small population size.”Please go to P19, L357-359.

Comment 13: On the line 387, Why are you talking about unbalanced sex ratio here. it comes from nowhere. I think it should be a new paragraph, and for your reader to follow you, remind them of the sex ratio you found (assuming these individuals were indeed males or not).

Response: We agreed your comments on this paragraph which makes it more logical and coherence. So, we started a new paragraph to talk about sex ratio and have rewritten some sentences. Please go to P20, L378-387.

Comment 14: On the line 390-394, This sentence makes no sense. There are two informations here, but how they are related is a mystery.

Response: This sentence has been deleted. 

Comment 15: On the line 399, Ok, it was observed in another population, but what about another one? You cannot cherrypick information as if were supporting yours. Is there a pattern among small population in crowded places? or in areas with low habitat quality? or anything that could explain why females exhibit this low body condition?

Response: The reference we cited before had similar situation on population density, but we had smaller study area. So, this sentence has been deleted. 

Comment 16: On the line 401, The comparison is male vs female in adult individuals: leave the immature out.

Response: The “immature males” has been deleted in this sentence. Please go to P21, L392. 

Comment 17: On the line 403, You surely mean collecting, which you already use.

Response: Actually, it should be grazing, we used the wrong word. Please go to P21, L394. 

Comment 18: On the line 419, Use a proper way to cite this. If these are unpublished documents, cite the staff who provided the information as personal communication. Also, a documentary is not documentation.

Response: This citation way has been changed to “Li Zhimin, personal communication”. Please go to P21, L411. 

Comment 19: On the line 437, Sorry, but how documenting this population will ensure its survival? It seems rather a leap of faith!

Response: Sorry the way we wrote the sentence confused you. This sentence has been changed to “The long-term placement of cameras in NNNR is an efficient mean of monitoring to obtain valuable data on demography, fecundity and body condition variation across seasons and years.” Please go to P22, L429-431. 

Comment 20: On the line 442, Where was it confirmed first? I don't remember reading anything on that earlier in the manuscript

Response: Sorry for that we used the wrong word. This sentence has been changed to “The results of this study match those observed in earlier studies that providing more suitable habitat is urgently needed to ensure an adequate supply of food for this isolated population during the dry season.” Please go to P22, L433-435.

Comment 21: On the line 447, of what?

Response: This sentence has been changed to “The short-term objective to increase the survival quality of this small population should be to compensate for the food shortages the NNNR population during the dry season by establishing more food sources and improving the quality of existing one to make sure all the individuals have a good body condition.” Please go to P23, L438-441.

Comment 22: On the line 448, by establishing what?

Response: Please see the response of comment 21.

Comment 23: On the line 459-460, Do you mean carrying capacity? I don't think it makes sense, unless you start with that and use it to demosntrate that the population, under the current conditions, is not viable.

Response: We agreed your opinion after deeply considering, so, we deleted the “environmental capacity”. 

Comment 24: On the line 461-463, That recommendation makes no sense. First, there is probably inbreeding since quite long: you do not provide evidence that the connection with another population has been lost recently. Second, in the current state of affairs, it seems that ensuring a better body condition is more important than ensuring reproduction.

Response: We agreed that you mentioned this population facing the health problems. So, this sentence has been changed to “We emphasize the urgency of implementing these recommendations especially in ensuring a better body condition, which may improve the success rate of reproductivity in the long run.” Please go to P23, L452-454.

---

## [Decision Letter · Decision Letter 2]

19 Jan 2021

PONE-D-20-11590R2

Population structure and body condition assessment to inform conservation strategies for a small isolated Asian elephant population in southwest China

PLOS ONE

Dear Dr. Shi,

Thank you for submitting your manuscript to PLOS ONE. After careful consideration, we feel that it has merit but does not fully meet PLOS ONE’s publication criteria as it currently stands. Therefore, we invite you to submit a revised version of the manuscript that addresses the points raised during the review process. Check and follow the suggestions from reviewer #2 contained in a paragraph below and in the attached PDF file.

We look forward to receiving your revised manuscript.

Kind regards,

Paulo Corti, Ph.D.

Academic Editor

PLOS ONE

Review Comments to the Author

Reviewer #2: I commend you for the care you took addressing all my previous comments. The revision by a native speaker was obvious (except for a couple of sentences that were probably added or changed posterior to that revision). However, you will note a few further corrections, sometimes just a coma; these corrections are needed as they change the arguments you make. I have no further comments regarding the interpretation of the data, except in the last paragraph, but this is easily fixed by ommitting the causal link you suggest. All my corrections and comments are in the joint pdf file.

---

## [Author Response · Author response to Decision Letter 2]

28 Jan 2021

Dear reviewers and editor,

Special thanks to reviewers and editors for providing exhaustive and positive guidance and suggestions for this paper for so long, which made this paper more and more perfect than the initial version. In this minor revision, we combined two native English speakers’ polished manuscript to make this paper suitable for English standard writing. We hope this version meet PLOS ONE’s publication criteria. The following letter gives our response to the reviewer’s specific comments are requested in this Minor Revision.

General Comments of Reviewer #2:

Response: Thank you very much for your recommendation in this minor revision. We have tried our best to revise the manuscript according to your kind and construction comments and suggestions. We have followed your advice and found native English speakers to polish this paper again. All the small problems in this version were fixed this time. We sincerely hope that this revised manuscript has addressed all your comments and you will be satisfied.

Specific Comments of Reviewer #2:

Comment 1: Abstract: “I do not know if you are already at the limit of words permitted for the abstract, but if you have some space allowed, you should mention the particularly dire body condition of reproductive females.”

Response: The sentence had been changed to “The average BCS was 5.75 (n=12, range 2-9), with adult females scoring lower than adult males.” Please go to Abstract.

Comment 2: Instead of writing a very general sentence, you should probably mention your 3 recommendations. Don't forget that most readers don't go further than the abstract.

Response: This general sentence had been deleted. New sentence had been added like “We propose three plans to improve the survival of this population: improving the quality and quantity of food resources, removing fencing and establishing corridors between the east and west parts of Nangunhe reserve.” Please go to Abstract.

Comment 3: I have made a number of comments in the abstract that you can find above. The abstract is your selling card!

Response: We totally agreed your opinion and made the Abstract showed more information.

Comment 3: On Line 83, Here it is better to make one sentence of the two first sentences, because the first one is not related to the paragraph, which is really about BCS. That's what I suggest with these corrections.

Response: The sentence had been changed to “Effective conservation strategies for small, threatened populations need detailed baseline information on both the population size and its habitat. Population structure and density estimation are other essential features that can be used to study the ecology of wildlife population while, the Body Condition Scoring (BCS) is an index of an animal’s health that reflects habitat quality.” Please go to P4, L81-85.

Comment 3: On Line 99, Repeat the complete name here.

Response: The sentence had been changed to “Although it is a unique evolutionary unit in China, the isolated Nangunhe National Nature Reserve population has been poorly studied.” Please go to P5, L 97-98.

Comment 3: On line 110, This is a sentence that seems more suited for the conclusion than here.

Response: This sentence had been moved to Conservation recommendations part. Please go to P21, L415-416.

Comment 3: On Line 124, I think I already mentioned that previously. If you provide the values for the dry seaon and the rain season, it is no annual mean. The mean annual precipitation is 351.3 + 1983.3 = 2334.6 mm.

Response: The word “annual” had been deleted. Please go to P6, L122.

Comment 3: On Line 355, If you use "although" there needs to be an argument to counterbalance. There no such argument in that sentence and it is in fact the following sentence. Because this first sentence is already log, I propose you make the changes I suggest in the text.

Response: The sentence had been changed to “The juveniles in this population accounted for 33% of the population, higher than the minimum of 20% needed for the population to be self-sustaining over a short period.” Please go to P18, L345-346.

Comment 3: On Line 379-382, loss of genetic diversity is far from being the most common cause of extinction in small populations!

Response: The sentence had been changed to “Male elephants can recognise kin and avoid inbreeding, and so male-biased populations usually indicate an increased genetic diversity and an ability to persist. However, without introductions from elsewhere, such a small population with only 7 males is likely to go extinct in the near future” Please go to P19, L363-366.

Comment 3: On line 438, The sentence was a mess, but also you have nothing to assert that there is some food shortage. You know that BCS tends to be lower, which might be indicative of that, but no proof. However, it makes perfect sense, in a management strategy, to try and provide more food.

Response: The sentence had been changed to “To improve elephant survival in the short term, we recommend establishing more food sources and improving the quality of existing sources to increase individuals’ body condition.” Please go to P22, L424-426.

---

## [Editor Report · Decision Letter 3]

23 Feb 2021

Assessing population structure and body condition to inform conservation strategies for a small isolated Asian elephant (Elephas maximus) population in southwest China

PONE-D-20-11590R3

Dear Dr. Shi,

We’re pleased to inform you that your manuscript has been judged scientifically suitable for publication and will be formally accepted for publication once it meets all outstanding technical requirements.

Kind regards,

Paulo Corti, Ph.D.

Academic Editor

PLOS ONE

---

## [Editor Report · Acceptance letter]

25 Feb 2021

PONE-D-20-11590R3 

Assessing population structure and body condition to inform conservation strategies for a small isolated Asian elephant (*Elephas maximus*) population in southwest China 

Dear Dr. Shi:

I'm pleased to inform you that your manuscript has been deemed suitable for publication in PLOS ONE. Congratulations! Your manuscript is now with our production department. 

Kind regards, 

on behalf of

Dr. Paulo Corti 

Academic Editor

PLOS ONE